# Hadamard Domain Training with Integers for Class Incremental Quantized Learning

## Abstract

Continual learning is a desirable feature in many modern machine learning applications, which allows in-field adaptation and updating, ranging from accommodating distribution shift, to fine-tuning, and to learning new tasks. For applications with privacy and low latency requirements, the computational and memory demands imposed by continual learning can be cost-prohibitive for resource-constraint edge platforms. Reducing computational precision through fully quantized training (FQT) simultaneously reduces memory footprint and increases compute efficiency for both training and inference. However, aggressive quantization especially integer FQT typically degrades model accuracy to unacceptable levels. In this paper, we propose a technique that leverages inexpensive Hadamard transforms to enable low-precision training with only integer matrix multiplications. We further determine which tensors need stochastic rounding and propose tiled matrix multiplication to enable low-bit width accumulators. We demonstrate the effectiveness of our technique on several human activity recognition (HAR) datasets and CIFAR100 in a class incremental learning setting. We achieve less than 0.5% and 3% accuracy degradation while we quantize all matrix multiplications inputs down to 4-bits with 8-bit accumulators.

## 1 Introduction

Humans have a remarkable ability to continuously acquire new information throughout their lives. Given neural networks (NN) this capability can make them more versatile and useful, as tasks can change over time, and data distributions can shift. Even when a task remains constant, the integration of new data is crucial for improving predictive outcomes. To accomplish this, NNs must strike a delicate balance, being adaptable enough to accommodate new tasks and data, while also remaining stable enough to preserve previously acquired knowledge. This inherent tension between adaptability and stability is referred to as the "Stability-Plasticity Dilemma" (Mermillod et al., 2013), and is the subject of Continual Learning (CL), Lifelong Learning, or Incremental Learning. A neural networks' inability to acquire new tasks without compromising their performance on previously learned tasks is commonly referred to as "Catastrophic Forgetting" (CF). Addressing CF stands as a significant obstacle in the pursuit of more capable autonomous agents, as highlighted by Mc-Closkey & Cohen (1989); Zhou et al. (2023). Techniques to mitigate the impact of CF include preventing information loss through regularized parameter updates, memory replay, or dynamic architectures (Parisi et al., 2018). Harun et al. (2023a) found that many popular continual learning algorithms are resource-expensive, with some CL algorithms being more costly than training the same model from scratch. R1/R2: Additionally Verwimp et al. (2023) found that especially compute costs are not discussed by many CL works. These costs are further exacerbated when additional constraints such as privacy, low-latency, or low-energy are imposed. R1/R2: Such constraints are typically encountered in edge devices in applications of HAR in mobile/AR/VR scenarios, drones, and miniaturized autonomous robots.

Operating with reduced precision tensors for an NN through quantization simultaneously increases energy-efficiency of computation and reduces memory footprint and bandwidth requirements. These improvements can be leveraged by custom compute primitives typically present in ML accelerators (Choquette, 2023), custom field programmable gate arrays (Jain et al., 2020), and custom integrated circuits (Garofalo et al., 2022). Additionally, since quantization can be applied together with

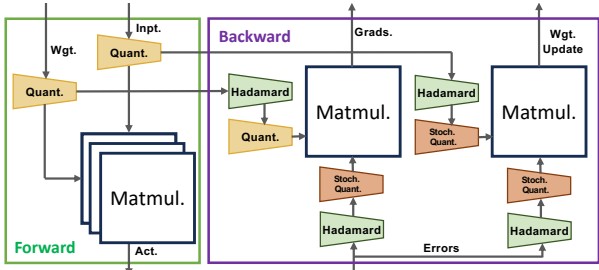

Figure 1: Overview of our proposed methodology, illustrating Hadamard transform and stochastic rounding in the backward pass and tiled matrix multiplication in the forward pass to ease the burden of the forward-pass accumulator.

any model architectural improvements, gains from quantization can typically be multiplicative with any architectural improvements. Previous work on fully quantized training (FQT), *e.g.,* accelerating matrix multiplication through quantization in forward- and backward-pass of the backpropagation algorithm, have shown promising results in non-CL scenarios (NoCL) through the use of custom floating point formats, stochastic rounding, or adaptive scales (Chmiel et al., 2023; Lee et al., 2022; Xi et al., 2023; Sun et al., 2020).

To the best of our knowledge, there has been little work in using FQT for CL to enable its application in resource-constrained environments. This naturally leads to the following question, "Can FQT be applied for CL and to what extent are they compatible?" Answering these questions is the focus of this paper. To do so, we introduce a new quantization algorithm for fully quantized CL training leveraging inexpensive Hadamard domain computation which we call Hadamard Domain Quantized Training (HDQT). Our main contribution are:

- We propose a quantized training technique where the backward-pass matrix multiplications are implemented in the Hadamard domain to make efficient use of quantization ranges (overview in Figure 1). We additionally determine which tensors in the backward-pass are negatively affecting learning and require stochastic rounding. We propose tile-based matrix multiplication for the forward pass to enable low-bit width accumulation. Thus, enabling training with standard 4-bit integers as inputs and 8-bit accumulators.

- We provide the first overview of the relationship between fully quantized training and continual learning with the example of class-incremental learning. We see at most a general accuracy reduction effect of quantization, however, we find no additional amplification of accuracy losses over the course of CIL training. Thus, indicating that FQT is a readily available tool for continual learning in resource-constrained environments.

- We evaluate our approach with three well-known continual learning techniques on the example of CIFAR100 a classic continual learning benchmark as well as three popular human activity recognition datasets. For a 4-bit training with 8-bit accumulators we find at best, -2.47%, -0.09%, +0.25% and -0.42% in the final accuracy changes for these datasets over the investigated algorithms compared to a floating point 16 baseline. For CIFAR100 if we increase the accumulator bit-width to 12 while keeping input bit-width constant we even get a 1.00% performance boost over the floating point implementation.

## 2 BACKGROUND

### 2.1 CLASS INCREMENTAL LEARNING (CIL)

In continual learning, CF is predominantly addressed by three classes of techniques: *regularization*, *memory replay*, and *dynamic architectures* (Parisi et al., 2018). Regularization terms such as knowledge distillation (KD) (Hinton et al., 2015) are used to penalize large parameter updates to mitigate forgetting. *Learning without forgetting* (LwF) (Li & Hoiem, 2016) is the earliest example, which employs KD as a regularization term in a task-incremental learning scenario. Memory replay approaches periodically update the network with the combination of the current training set with holdout data samples subsampled from previous tasks' training data (Hetherington & Seidenberg,

1989; Robins, 1993; 1995). Representative examples include *incremental classifier and representation learning* (iCaRL) (Rebuffi et al., 2017), *maximal interfered retrieval* (MIR) (Aljundi et al., 2019), *learning a unified classifier incrementally via rebalancing* (LUCIR) (Hou et al., 2019), and *look-ahead MAML* (La-MAML) (Gupta et al., 2020). Generative techniques have also been studied to augment or replace holdout data samples; *e.g., replay through feedback* (RtF) (van de Ven et al., 2020). Dynamic architecture approaches extend or change the network's architecture to increase the capacity of the network for new tasks. These techniques can grow the network as with *progressive networks* (Rusu et al., 2016) and *dynamically expandable representation* (DER) (Yan et al., 2021) or isolate and freeze parameters like *packnet* (Mallya & Lazebnik, 2018).

Recent research indicates that CF is largely caused by a bias in the weights towards new classes. This has spawned a new area trying to remove this bias and thereby decreasing forgetting. Notable examples are *bias correction* (BiC) (Wu et al., 2019), *weight alignment* (WA) (Zhao et al., 2019), or *vector normalization and rescaling* (WVN-RS) (Kim & Kim, 2020).

Continual learning has attracted increasing interest within the human activity recognition (HAR) community (Ye et al., 2019). Jha et al. (2021) have conducted an empirical assessment on a wide array of state-of-the-art continual learning techniques to HAR datasets. The results have demonstrated promising performance to support incremental learning on new activities. Leite & Xiao (2022) have proposed an efficient strategy for HAR that leverages expandable networks, which dynamically grow with the incoming new classes. They utilize the replay of compressed samples selected for their maximal variability. Zhang et al. (2022) have explored multimodal learning in a continuous fashion for HAR. Their approach combines EWC (Kirkpatrick et al., 2017) with canonical correlation analysis to harness correlations among various activity sensors, with the aim of mitigating catastrophic forgetting. Schiemer et al. (2023) present an online continual learning scenario for HAR where their technique *OCL-HAR* has to perform real-time inference on streaming sensor data while simultaneously discovering, labelling and learning new classes.

There have been several studies on improving the efficiency of continual learning. Hayes & Kanan (2022) define criteria for CL algorithms on the edge and evaluate popular models' performance using CNNs designed for the edge. Wang et al. (2022) create SparCL a framework for continual learning on the edge. They leverage weight sparsity, effective data selection and gradient sparsity to reduce the computational demand achieving $23\times$ less training FLOPs while improving SOTA results.

To the best of our knowledge, this paper presents the first work that uses FQT in conjunction with CL, while previous connections between quantization and continual learning have been focusing mainly on weight quantization and feature compression for replay memory. Shi et al. (2021) introduce weight quantization to CL through bit-level information preserving, where they quantize to 20-bits and use quantization behavior as a signal to identify important task parameters. Pietroń et al. (2023) use weight quantization to create efficient task sub-networks. Ravaglia et al. (2021) employ quantized latent replay through 7-bit quantization of the frozen old model, prototyping their idea on system-on-a-chip (VEGA). They yield a $65\times$ computational speed increase while being $37\times$ more energy efficient compared to an STM32 L4 baseline. The most prominent application is not parameter quantization but feature compression (vector quantization) using autoencoders to optimize sample memory storage; e.g. Hayes et al. (2019); Caccia et al. (2020); Srivastava et al. (2021); Wang et al. (2021); Harun et al. (2023b). While vector quantization certainly improves learning efficiency, it is not a holistic resource optimization approach.

## 2.2 FULLY QUANTIZED TRAINING (FQT)

There has been a renewed interest in quantized learning due to the increasing costs associated with training larger and more capable models. Several techniques have seen widespread use including specific floating-point formats, adaptive gradient-scaling, modified rounding schemes for quantization, amongst others. We provide a short overview of some of the related work here, and a more extensive overview in the Appendix A.1.

Different floating-point representation formats, such as the radix-4 format introduced in Sun et al. (2020) have seen wide study including recent work on FP8 training with commodity hardware van Baalen et al. (2023). The authors in Sun et al. (2020) also include an adaptive scaling scheme with layerwise updates of the gradient scales to minimize value clipping. They additionally employ a multi-phase rounding scheme for their weights and error gradients to effectively increase resolution

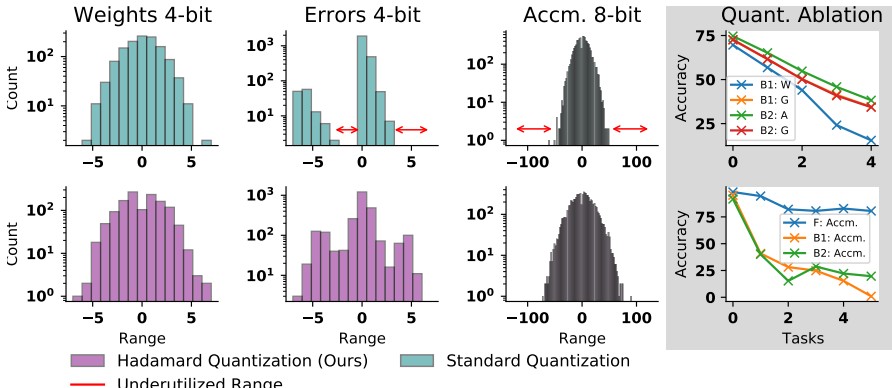

Figure 2: Comparing Hadamard domain quantized matrix-multiply with quantized matrix-multiply for 4-bit inputs (weight and errors) and 8-bit accumulation. The Hadamard domain operations make better use of the quantization range, with fewer empty bins. There is consistently more efficient use of the quantization levels for all tensors, with the difference accentuated for error tensors and the output of the matrix-multiply (Accm.). On the right R4: grey background, we show results from an ablation study for quantizing tensors in the backward pass (top) and accumulator quantization (bottom). The model accuracy is most sensitive to quantizing the gradients and activation-tensor. We also observe that not quantizing the accumulator in the forward pass has an outsized impact on accuracy, compared to not quantizing the accumulators in the backward pass.

leading to a less than 1% drop in accuracy on ImageNet and several NLP tasks. Other efforts include, Chmiel et al. (2020), who achieve iso-accuracy for 6-bit quantized gradients on ImageNet, by leveraging the distribution differences between forward pass tensors and gradients. By modelling the distribution of gradients as log-normal, the authors derive specific values for the mantissa and exponent in their floating-point format. They then build on these results in, Chmiel et al. (2023) to develop an exponent-only format for the gradients using stochastic rounding to unbias gradient estimation. This delivers accuracy within 1.1% of a floating-point ResNet50 for 4-bit quantization.

Lee et al. (2022) developed a principled approach to determine a good floating point format (number of exponent and mantissa bits). They examine the misalignment in weight gradient errors compared to a high-precision backward pass and introduce quantization hysteresis. This results in a 4-bit weight, 8-bit activation and 8-bit gradient ResNet18 training to within 0.2% of their floating-point baseline accuracy. FQT has also been examined for language models with Xi et al. (2023) proposing to use a Hadamard transform during the forward pass to better capture outlier values in the activations with bit-splitting in the backward-pass to speedup training performance on GPUs by 35.1%.

## 3 METHODOLOGY

CL must accommodate changing distributions in data, weights and gradients, additionally, research indicates that forgetting can be very sensitive to small perturbations in a few parameters (Zhao et al., 2023). To address these challenges in CL, we propose a lightweight FQT approach which avoids constant reconfiguration typical in FQT and employs uniform integer quantization. Our proposed FQT approach enables matrix multiplication of the form $\mathbf{Z} = \mathbf{X}\mathbf{W}$, where $\mathbf{Z} \in \mathbb{R}^{N \times C}$, $\mathbf{X} \in \mathbb{R}^{N \times D}$ and $\mathbf{W} \in \mathbb{R}^{D \times C}$. These operations are required for both forward- and backward-pass. We emulate hardware quantization in software as $Q(\mathbf{x}) = \text{round}(\text{clip}(\alpha \cdot \mathbf{x}) \cdot 2^{b-1}) \cdot 2^{-(b-1)} \cdot \alpha^{-1}$. Where $Q$ is the quantization function which takes in a high precision value $x$ and converts it to a $b$-bit integer value. Additionally, $\alpha$ scales the values to lie between $\pm 1$, clipping saturated values outside this range, and rounding returns the nearest integer. We determine $\alpha$ by a calibration technique based on the maximum value in that tensor. During training, most tensors especially the gradients vary over a wide dynamic range, requiring an adaptive quantization scheme to be used for FQT.

Typically, the wide dynamic range of tensors is addressed through custom floating-point formats that trade-off precision for an improved representable range (Lee et al., 2022). This aligns with observations indicating that gradient distributions exhibit power-law heavy tails (Xie et al., 2022).

However, for a given number of bits, floating-point operations are typically more expensive than the equivalent integer operation. Instead, we avoid the precision-dynamic range trade-off and operate only with integers by implementing computations in a transform domain. Because the Hadamard transform[1] is an energy-preserving, orthogonal, linear transform that can be implemented using only $n \log(n)$ additions, it is a natural choice for our work (Xi et al., 2023). The information-spreading property of the Hadamard transform allows us to make better use of all available quantization levels, capturing information from the tails of the distribution. The Sylvester form (Sylvester, 1867) of the Hadamard transforms is defined as R4: $\mathbf{H}_0 = [1], \; \mathbf{H}_k = \frac{1}{\sqrt{2}} \left[ \mathbf{H}_{k-1} \quad \mathbf{H}_{k-1}; \; \mathbf{H}_{k-1} \quad -\mathbf{H}_{k-1} \right]$.

Hadamard matrices are orthogonal and symmetric $\mathbf{H}_k = \mathbf{H}_k^T = \mathbf{H}_k^{-1}$, so $\mathbf{H}_k \mathbf{H}_k = k\mathbf{I}$. As a result, $\mathbf{X}\mathbf{H}\mathbf{H}^{-1}\mathbf{W} = k\mathbf{X}\mathbf{W}$, with the Hadamard and it's inverse becoming $k\mathbf{I}$. Due to this property, the products of Hadamard-transformed tensors need not be transformed back from the Hadamard domain. Since Hadamard matrices are only defined for powers of two, we use block-diagonal transformation matrices $\mathbf{H} \in \mathbb{R}^{D \times D} : \text{BlockDiag}(\mathbf{H}_k, ..., \mathbf{H}_k)$. Here, the highest $k$ is chosen to deliver a valid block-diagonal matrix for the given tensor dimension (*e.g.,* dimensions of $\mathbf{X}$ or $\mathbf{W}$). Figure 2 compares the distribution of quantized tensors, with and without the Hadamard transform. As seen by the underutilized bins (see red arrows) when directly quantizing the errors to 4-bits, the Hadamard transform is more efficient in using available quantization levels. We also observe consistently that this spreading also causes the results from Hadamard domain matrix multiplication to better use the full dynamic range available for accumulation (8-bits shown in Fig. 2).

To improve performance, we study its sensitivity to quantizing different tensors in the backward pass. Figure 2 (right) shows an ablation study where we quantize all but one tensor. We see significant improvements when the two gradient tensors and the activation tensor in the backward pass are unquantized. To mitigate the impact of bias introduced due to quantization in these sensitive tensors, we use stochastic rounding (Chmiel et al., 2023; Liu et al., 2022). Figure 2 (right, bottom) shows results from an ablation study where we do not quantize the accumulator of the matrix multiply units. Our results indicate that accumulating the partial sums during the forward pass at a lower precision adversely affects accuracy. We mitigate this through tiled quantization, which reduces the partial sums, limiting computations to tiles of size $s$ R1: (similar to (Wang et al., 2018b)). We hypothesize that gradient sparsity prevents an impact on the backward pass (see Figure 2).

## 4 EXPERIMENTS

We focus on supervised class incremental learning where each incoming task has disjoint classes and task identities are unknown during testing. Let $T = [t_1, t_2, \ldots, t_n]$ be a sequence of tasks where each task $t_i$ holds a finite set of classes $C_i$ and their $n_i$ training samples $\{(x^j, y^j)|y^j \in C_i\}_{j=1}^{n_i}$ are disjoint from previous tasks $t_k < t_i$, with $k \in 1, \ldots, i-1$. The goal is to integrate new information into an existing model without regressing performance on prior tasks. We do not require any task IDs or further knowledge. Thus, our method is broadly applicable to different CL scenarios *e.g.,* online Continual Learning.

To evaluate our performance we choose average class accuracies $a_c$ for each task and the forgetting measure. The forgetting score by Chaudhry et al. (2018) measures the difference between the best and current accuracies, indicating how much knowledge of previous tasks has been lost when learning new tasks. Forgetting at task $t_i$ is defined as: $F_i = \frac{1}{|C|} \sum_{c=1}^{C} f_c$ where $f_c = \max_{l \in 1, \ldots, i-1} (a_c^l) - a_c^i$.

We evaluate HDQT on CIFAR100 (Krizhevsky, 2009) and three popular HAR datasets: PAMAP2, DSADS, and HAPT Jha et al. (2021); Schiemer et al. (2023). These datasets have been commonly used as a baseline to evaluate CIL algorithms Rebuffi et al. (2017); Wu et al. (2019) and a brief introduction of HAR datasets can be found in Appendix A.3. We study the interaction between FQT and continual learning through 3 representative CIL techniques: LwF (Li & Hoiem, 2016), iCaRL (Rebuffi et al., 2017) and BiC (Wu et al., 2019) with the implementations from Zhou et al. (2023)[2]. Appendix A.2 and A.10 presents the description of each technique and additional results.

For CIFAR100, we use ResNet-32 with the same set of hyperparameters as in Zhou et al. (2023). For HAR datasets, we choose three layered fully connected networks (FCN) where each layer has the same amount of neurons as the input layer. The output layer has as many neurons as there are

---

[1]R3: For a detailed introduction to hadamard transforms please consult Agaian et al. (2011).

[2]https://github.com/zhoudw-zdw/CIL_Survey

Table 1: Results for standard learning (NoCL) and CIL with 20 classes (CIFAR100) or 2 classes (DSADS, PAMPA2, or HAPT) per task quantized (4-bit inputs and 8-bit accumulators) and unquantized (FP). Numbers are final accuracy percentages averaged over 20 runs of the HAR datasets and 5 runs for CIFAR100 (small numbers in brackets are standard deviation). Note that for quantized CIL on HAR datasets, iCaRL is the best CIL method and for CIFAR100 BiC (see underlined numbers).

| | CIFAR100 | | DSADS | | PAMAP2 | | HAPT | | |
| | FP | 4-bit | FP | 4-bit | FP | 4-bit | FP | 4-bit | $\mathbb{E}[\Delta\text{Acc.}]$ |
|---|---|---|---|---|---|---|---|---|---|
| NoCL | 68.61 | 59.43 | 84.44 | 82.71 | 85.81 | 84.35 | 93.15 | 92.08 | *3.36* |
| | (±0.24) | (±1.88) | (±4.36) | (±4.45) | (±4.91) | (±7.66) | (±2.11) | (±2.10) | |
| LwF | 33.92 | 29.32 | 12.06 | 12.33 | 36.34 | 32.84 | 32.78 | 28.65 | *2.99* |
| | (±0.91) | (±1.28) | (±6.01) | (±7.68) | (±8.03) | (±10.25) | (±8.74) | (±10.22) | |
| iCaRL | 43.20 | 37.25 | 72.83 | **72.74** | 77.52 | **77.87** | 82.98 | **82.56** | *1.53* |
| | (±1.21) | (±2.42) | (±4.84) | (±4.59) | (±5.50) | (±5.02) | (±5.12) | (±4.51) | |
| BiC | 48.54 | **46.07** | 67.55 | 63.79 | 77.04 | 75.46 | 81.42 | 78.26 | *2.74* |
| | (±3.95) | (±2.64) | (±6.05) | (±5.86) | (±7.00) | (±5.79) | (±4.32) | (±5.14) | |
| $\mathbb{E}[\Delta\text{Acc.}]$ | *5.69* | | *1.33* | | *1.55* | | *2.20* | | |

currently seen classes and has to be extended when a new task comes in. HDQT introduces two quantization-specific hyperparameters: a block-size for accumulators (32) and a scaling factor on the maximum to remove outliers before integer quantization in the forward pass for weights and activations (0.975). Additional hyperparameter settings are outlined in Appendix A.5.

## 4.1 RESULTS

Table 1 summarizes the final accuracy of 4-bit quantized models and FP models. HAR datasets with quantized iCaRL incur minimal accuracy degradation, with the largest accuracy drop observed to be 0.42% on HAPT. BiC achieves the best performance on CIFAR100 with and without quantization, consistent with previous results (Wu et al., 2019). Quantization effects are more pronounced for CIFAR100 (-2.47% for BiC) vs. HAR datasets (DSADS -0.09%, PAMAP2 +0.35%, and HAPT -0.42% for iCaRL). We observe a lower loss in accuracy from quantization when using CIL compared to models trained with NoCL (-3.36% NoCL vs. -2.99% LwF). The standard deviation in model accuracy for CIL through HDQT is similar to what is achieved on FP-trained models, indicating minimal impact in variance for HDQT.

Figure 3 shows average class accuracy and forgetting per step using 4-bit weights and 8-bit accumulation for HDQT and the baseline FP model. HDQT impacts the accuracy of the initial model for CIFAR100, with a drop of 7.08% on BiC at Task 0. However, the per-task learning trajectory for different CL techniques follows a similar trend for quantized and unquantized models across datasets. The correlation coefficient between FP and HDQT is consistently high (0.99–1.00) on the averaged trajectories, suggesting that CIL and HDQT together do not further degrade model accuracy.

In Figure 4 we show the effect of varying input bit-widths for all three CIL techniques on CIFAR100 while keeping the accumulator bit-widths as double of the input bit-width to account for 'bit-growth'[3] during accumulation. For all techniques, we observe that quantizing inputs to 3-bits causes a severe drop in accuracy (*e.g.,* -19.7% on CIFAR100 with BiC compared to 4-bits). Input precision higher than 4-bits shows no additional benefit. The ablation study demonstrates how much bit-growth occurs during accumulation. We fix the input bit-width at four and observed accuracy over multiple settings of accumulator-bit-width. When constrained to 4-bits, HDQT fails, with accuracy no better than random. Increasing accumulator bit-width over our default of 8-bits (12- and 16-bits) leads to only ∼3% improvement in the final accuracy, suggesting a favorable trade-off in precision and efficiency at 8-bits. We observe that different degrees of quantization still result in

---

[3]Bit-growth occurs when mathematical operations (e.g., matrix multiplications) increase a result's bit-length beyond a system's representational limit.

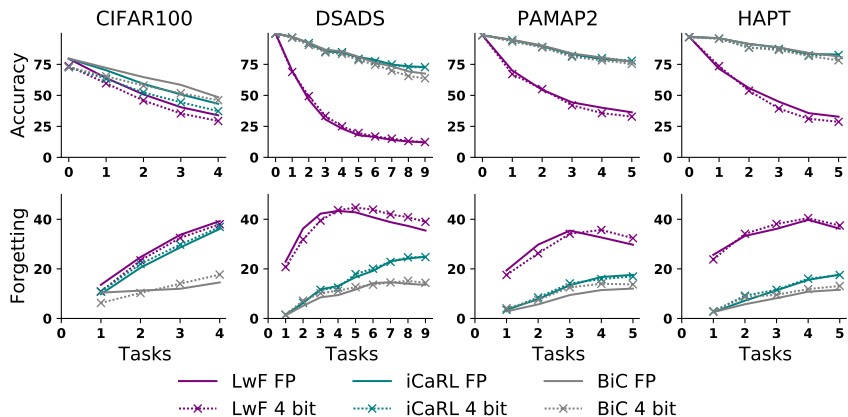

Figure 3: Step accuracy (first row) and forgetting (second row) of quantized (matrix multiplications in forward and backward-pass with 4-bit inputs and 8-bit accumulators) and FP models for CIFAR100, DSADS, PAMAP2, and HAPT datasets averaged over 5 (CIFAR100) and 20 (HAR datasets) runs with different class orders. Note the intersection of the forgetting lines of quantized and FP models for the BiC on CIFAR100, LwF on DSADS and PAMAP2, indicate model learning capacity saturation mid-training in contrast to other models.

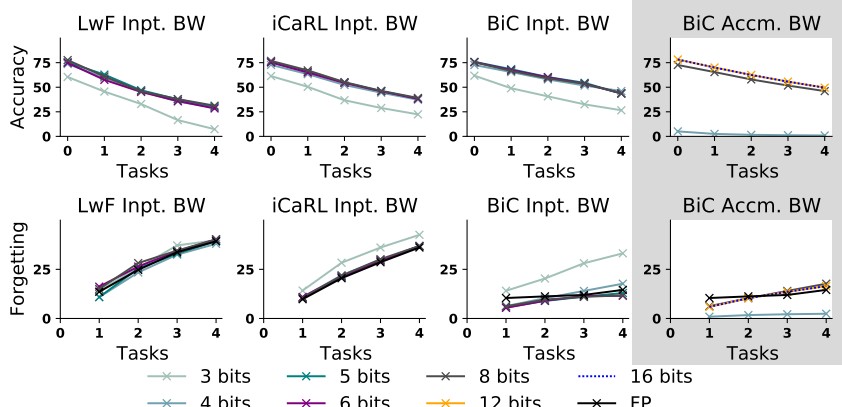

Figure 4: Step accuracy averaged over three runs on CIFAR100 for LwF, iCaRL and BiC with different bit widths ranging from three, four, five, six, and eight for inputs and four, eight, twelve and sixteen for accumulators R4: (grey background).

similar learning over different tasks (similar change in accuracy per step). Again this confirms that HDQT reduces initial training accuracy, but minimally impacts CIL.

Despite initial accuracy variations, the training curves of FP and HDQT models display strong similarity, indicating comparable learning performance. However, distinct forgetting trajectories reveal changes in representational quality at each incremental step. By examining both models' learning and forgetting patterns, qualitative differences in features learned through FQT become apparent. The consistent change in accuracy between tasks (correlation of 0.99–1.00) for HDQT and FP implies they learn at the same rate.

To better anaylze their difference, we categorize forgetting behaviors into distinct scenarios. Scenario 1: the quantized model forgets more than the unquantized model, indicating that the quantized model has reached its representational limit. Scenario 2: the quantized model forgets less, indicating it has not yet reached its representational limit. We analyze forgetting curves in Fig. 3 to further understand how FQT on CIL algorithms impacts the representational limit. Consider LwF on CIFAR100, since the FP model always forgets more than the HDQT model hinting that the quantized model has not yet reached its representational limit. On the other hand, if we consider in the same

figure iCaRL for CIFAR100, the HDQT model always forgets more than the FP model, indicating it is already at its representational limit. When the forgetting curves intersect, we can infer that the HDQT model has reached its representational limit (*e.g.,* see BiC on Task 3 for CIFAR100, suggesting that the learning technique has reached its representational limit for the quantized model during that task). When compared to previous training steps, from there on the algorithm may have to replace its previous knowledge with new knowledge to learn a new task at a higher rate when compared to the baseline FP CIL process.

We see the capacity changes over different bit-widths in Figure 4 where the crossover points between the FP model and the quantized models shift towards the right (especially visibly for BiC) given more bits and therefore indicating a higher representational limit for the model.

Compared to CIFAR100, the HAR datasets show a smaller forgetting gap between the FP and HDQT models. As shown in Table 1, HDQT on HAR datasets has negligible performance impact when using iCaRL, indicating that iCaRL's 4-bit network poses sufficient learning capacity to produce on-par results to floating point models. The results we saw in CIFAR100 for iCaRL and BiC are also reflected in the HAR datasets with BiC and LwF. We observe that HDQT saturates earlier (*e.g.,* see the forgetting curve of LwF on DSADS at task 4), indicating that new knowledge may have replaced old knowledge, leading to a higher average forgetting than the FP equivalent. LwF's low performance is consistent with previous research that has shown that regularization alone is not enough to mitigate forgetting (Knoblauch et al., 2020; Jha et al., 2021).

## 4.2 Effects of Forgetting from Quantization and CIL

Similar to Hooker et al. (2019), we study how HDQT affects forgetting in CIL. Figure 5 shows class-specific changes in the accuracy of quantization and continual learning on a single example run on the DSADS dataset. On quantizing a NoCL model directly using HDQT, we observe a minimal change in accuracy of 1.73%. Notably, CIL causes a greater total accuracy reduction of 11.61%. We do not observe any consistent change in accuracy due to HDQT applied with CIL. Although there are examples where quantization and CIL combined amplify forgetting, *e.g.,* A (Sitting) or D (Lying - Right Side), there are others where the opposite effect is observed , *e.g.,* C (Lying - Back). On comparing the results for CIL with those for CIL combined with HDQT, there is no distinct trend in forgetting observed for earlier classes in the training sequence (I, S, J ... ), indicating that HDQT does not amplify forgetting of the earlier classes.

## 4.3 Comparison to Other Quantization Work

To the authors' knowledge, there is no other work applying FQT to CIL. Thus, we ported two competitive FQT techniques, logarithmic unbiased quantization (LUQ) (Chmiel et al., 2023) and FP134 (Lee et al., 2022), into our CIL framework to enable a comparative study. Table 2 shows the resulting final accuracies of these two algorithms compared to HDQT with an 8- and 12-bit accumulator. LUQ uses 4-bit integer inputs to matrix multiplications in the forward pass and a specific 4-bit radix-2 format for the gradients with no accumulator quantization. Our method uses a standard 4-bit integer as input with 8-bit integer accumulators in forward and backward-pass. For fair comparison, we also include an 8/R2: 12-bit accumulator in the LUQ forward pass, which resulted in numerical instabilities, preventing learning. The authors of Lee et al. (2022) primarily use 8-bit floating point formats, and in addition, they include results from FQT with 4-bit floating point weights. We include both techniques in Table 2 and we additionally also include their results with all inputs reduced to 4-bit floating point. However, this resulted in numerical instability, leading to the model performance being no different from random. Given an increased bit-width for the accumulator from 8- to 12 as well as 16-bits our method outperforms others (including our floating point baseline) when using BiC training. R1: R2: In appendix A.14 we investigate reasons for our better performance at lower bit-widths. We find that HDQT delivers better gradient approximates.

## 4.4 Energy Considerations

Table 3 lists performance estimates for running HDQT with different numerical precision for the hidden and output layers for HAPT training. These were derived using established architectural performance estimation tools (Wu et al., 2019; Parashar et al., 2019) calibrated to a commercially available FinFET process. We use the architectural cost model for Nvidia's SIMBA (Shao et al., 2019) accelerator configured to 32 KB weight buffer, 8 kB input buffer, 64 kB global buffer with

Table 2: Final accuracy and forgetting values for all three CIL techniques using LUQ (Chmiel et al., 2023), FP134 (Lee et al., 2022), and our method given 8- and 12-bit accumulators on CIFAR100 averaged over five runs. $^\ddagger$ indicates that we only took the average over four runs because one run resulted in random accuracy due to numerical issues. $^*$Denotes when we use a modification to the LUQ code (see Section A.9 in the Appendix for detailed discussion). § denotes at least 1 failed run omitted from result.

| | Inpt. | Accm. | R4: NoCL | LwF | | iCaRL | | BiC | |
|---|---|---|---|---|---|---|---|---|---|
| | BW | BW | Acc.↑ | Acc.↑ | Forg. ↓ | Acc.↑ | Forg. ↓ | Acc. ↑ | Forg. ↓ |
| Base | 16 | 16 | 68.61 | 33.92 | 39.25 | 43.20 | 36.10 | 48.54 | 14.53 |
| LUQ | 4 | 16 | 65.30 | 33.23 | 36.56 | 42.46 | 35.65 | 48.58 | 13.58 |
| LUQ$^*$ | 4 | 16 | 65.14 | 33.66 | 35.60 | 41.72 | 36.24 | 47.91 | 14.20 |
| R2: LUQ$^*$ | 4 | 12 | Fail | Fail | | Fail | | Fail | |
| LUQ$^*$ | 4 | 8 | Fail | Fail | | Fail | | Fail | |
| FP134 | 8 | 16 | 65.91§ | 35.55 | 25.60 | 38.80 | 35.35 | 48.43$^\ddagger$ | 13.35$^\ddagger$ |
| FP134 | 4/8 | 16 | 65.81§ | 35.15 | 24.70 | 38.99 | 35.65 | 47.49$^\ddagger$ | 13.86$^\ddagger$ |
| FP134 | 4 | 16 | Fail | Fail | | Fail | | Fail | |
| HDQT (ours) | 4 | 16 | 61.96 | 30.21 | 39.56 | 39.07 | 37.70 | 49.00 | 17.01 |
| HDQT (ours) | 4 | 12 | 62.31 | 29.90 | 40.00 | 39.26 | 37.80 | 49.54 | 16.47 |
| HDQT (ours) | 4 | 8 | 59.43 | 29.32 | 38.00 | 37.25 | 36.93 | 46.07 | 17.69 |

64 MAC units (8 lanes of 8 wide vector macs). When constraining the architecture to the same cycle-count, our estimations indicate a $4.5\times$ improvement in energy efficiency in implementing the hidden layer and $3.8\times$ for the output layer. These are driven by a $18.7\times$ reduction in forward pass energy for the hidden layer and $6.4\times$ in the output layer. The backward pass has a $3.3\times$ and $3.1\times$ lower energy consumption for the hidden and output layer respectively. For instance, applying this to LwF we have gains not only in the forward and backward pass but additionally we have gains in the inference step of the previous model used for KD. R1: We show up to $4.5\times$ energy savings when reducing accumulators precision from 16 bit, indicating a favourable tradeoff. The energy overhead of using the Hadamard Transform is tabulated in A.8. The limitations of our performance estimation software limit us to using a matrix-multiply formulation for the Hadamard transform rather than the recursive butterfly (see A.4). Energy estimates for other datasets are shown in Appendix A.11.

Table 3: Accelerator cost-model derived estimates for the energy incurred to compute two representative layers of the FCN for HAPT. We include all operations, *e.g.,* max, FHT, etc. With a reduction in accumulator bits, results indicate an improvement of up to $4.0\times$ over the FP16 baseline.

| Layer | Inpt. BW | Accm. BW | Inpt. Dim. | Wgt. Dim. | FWD Energy $\mu$J | BWD Energy $\mu$J | Total Energy $\mu$J | Improvement (vs FP16) |
|---|---|---|---|---|---|---|---|---|
| Hidden | FP16 | FP16 | $128 \times 561$ | $561 \times 561$ | 1630.6 | 3319.8 | 4950.4 | 1.0 |
| | 4 | 16 | $128 \times 561$ | $561 \times 561$ | 99.6 | 1412.5 | 1512.1 | 3.3 |
| | 4 | 12 | $128 \times 561$ | $561 \times 561$ | 93.6 | 1358.8 | 1452.4 | 3.4 |
| | 4 | 8 | $128 \times 561$ | $561 \times 561$ | 87.0 | 991.3 | 1078.3 | 4.5 |
| Output | FP16 | FP16 | $128 \times 561$ | $11 \times 561$ | 45.0 | 88.9 | 133.8 | 1.0 |
| | 4 | 16 | $128 \times 561$ | $11 \times 561$ | 6.8 | 33.4 | 40.2 | 3.3 |
| | 4 | 12 | $128 \times 561$ | $11 \times 561$ | 6.6 | 30.6 | 37.2 | 3.6 |
| | 4 | 8 | $128 \times 561$ | $11 \times 561$ | 6.4 | 28.7 | 35.1 | 3.8 |

## 5 CONCLUSIONS

We introduce HDQT, an FQT quantization method which leverages the Hadamard domain to make efficient use of quantization ranges in the backward pass and apply it to CIL. We show promising results on image (CIFAR100) and HAR (DSADS, PAMAP2, HAPT) datasets, with a $\leq 2.5\%$ degradation through quantization on vision and $\ll 1\%$ for HAR with 4-bit inputs. When using 12-bit accumulation we beat our FP baseline on CIFAR100, indicating that we can exceed existing models by exploring the trade-offs between energy-efficiency and accuracy. This paves the road for learning continually on energy-efficient platforms.

## REPRODUCIBILITY STATEMENT

The code for our experiments can be found in the attached supplementary materials. We apply a standard pre-processing pipeline to CIFAR100 data, which normalizes each channel and during training applies random-crop, random-horizontal flips and color-jitter. For PAMAP2 the users 3, 4, 9 and class 24 have been removed, because the users do not have all classes and 24 is severely underrepresented. For HAPT's users 7 and 28 as well as class 8 the same has been done. No other steps have been performed. The exact pre-processing steps can be found in utils/data.py and utils/har_data.py. All our experiments have been seeded to ensure reproducibility and we provide bash files to start the experiments.

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

# A APPENDIX

## A.1 DETAILED FQT BACKGROUND

The authors in Chen et al. (2020) propose the idea that updates using FQT should align with updates from quantization-aware training (QAT), where weights and activations are quantized but the backward passes use floating point. They formulate FQT as a statistical estimator of QAT, capturing the performance loss due to the variance of the estimator. Building on this, they devise a variance minimization scheme using a 5-bit block Householder quantizer, resulting in a 0.5% accuracy loss on ResNet50.

Huang et al. (2022) highlight that especially quantization-aware training methods are costly because they require two copies of the weights and incur a lot of data movement. They propose resource constraint training which only keeps quantized weights which heavily reduces the burden for training and enables training on the edge. Their method adjusts the per-layer bit-width dynamically to train efficiently with just the right amount of bits necessary to facilitate learning. They show energy saving for general matrix multiplications of 86% and data movement while incurring less than a 1% accuracy penalty on various models such as BERT (SQuAD and GLUE) or ResNet18.

Most FQT techniques use dynamic quantization where ranges are computed on the fly given current data. Fournarakis & Nagel (2021) argue that this is very expensive and suggest that it is better to statically quantize data given range parameters estimated from previous training steps. They demonstrate minimal accuracy loss of MobileNets and ResNet18 while estimating a 50-400% memory cost difference between static quantization methods and their hindsight quantization.

Given the challenging nature of low-precision optimizers, e.g. keeping model parameters and gradient accumulators at low precision, Li et al. (2022) do a thorough analysis and construct an optimizer called low-precision model memory (LPMM). They claim that the biggest challenge is underflow and use stochastic rounding and microbatching to alleviate it. They use their optimizer to train up a ResNet18 and ResNet50 to SOTA with 70% reduced model memory.

Alternatively, Xu et al. (2022) analysis FQT for models with batch normalization. They find that batch normalization can amplify gradient noise. They add a rectification loss which reduces noise the bachtnorm noise amplification and show competitive results on 8, 4, and 2-bits.

In contrast, Liu et al. (2022) argue that the issue with quantization gradient noise is not variance but bias. Additionally, they claim that commonly used stochastic rounding can not prevent the "training crash" sometimes observed with fully quantized training. To enable FQT they propose a new adaptive quantization method called piecewise fixed point (PWF), which divides gradients into gradients close to zero and ones that with bigger magnitudes and both are quantized with different step sizes. They demonstrate that their method can enable 8-bit quantization for weights, activations, gradients and errors on ImageNet (image classification), COCO (vision detection), BLEU (machine translation), optical character recognition, and test classification while achieving $1.9 - 3.5\times$ speed up over full precision training with accuracy loss of less than $0.5\%$.

Another example for LLM training quantization is Yang et al. (2023) introduce dynamic stashing quantization, which quantizes operations and intermediate results to reduce DRAM traffic at the beginning with low precision and toward the end of the training the bit-width increases. They are able to reduce the number of arithmetic operations by $20.95\times$ and the number of DRAM operations by $2.55\times$ on IWSLT17.

## A.2 CONTINUAL LEARNING TECHNIQUE DETAILS

The techniques used in our investigation are representative of a variety of research directions and are often used as reliable baselines for continual learning. **Learning without Forgetting** Li & Hoiem (2016)'s objective is to maintain the similarity between the output of the old tasks in the new network to the output from the original model. This is achieved by using the KD loss as a regularization term. KD is a way to transfer knowledge from a complex *teacher* model to a simpler *student* model by minimizing the loss on the output class probabilities from the teacher model (Hinton et al., 2015). KD has found widespread application in various continual learning techniques to distill knowledge learned from previous tasks to the model for new tasks. It is defined as follows:

$$\mathcal{L}_{KD}(y_o, \hat{y}_o) = -\sum_{i=1}^{l} y_o^{'(i)} \log \hat{y}_o^{'(i)}, \tag{1}$$

where $l$ is the count of class labels, and $y_o^{'(i)}$ and $\hat{y}_o^{'(i)}$ are temperature-scaled *recorded* and *predicted* probabilities for the current sample for an old class label $i$. Temperature is used to address over-confident probabilities that the teacher model produces on their prediction. That is:

$$y_o^{'(i)} = \frac{(y_o^{(i)})^{1/T}}{\sum_j (y_o^{(j)})^{1/T}} \text{ and } \hat{y}_o^{'(i)} = \frac{(\hat{y}_o^{(i)})^{1/T}}{\sum_j (\hat{y}_o^{(j)})^{1/T}}.$$

The loss $\mathcal{L}_{KD}$ is combined with the cross-entropy loss on a new task's samples $\mathcal{L}_{CE}^n$; that is,

$$\mathcal{L}(y_n, \hat{y}_n, y_o, \hat{y}_o) = \lambda_o \mathcal{L}_{KD}(y_o, \hat{y}_o) + \mathcal{L}_{CE}(y_n, \hat{y}_n), \tag{2}$$
$$\mathcal{L}_{CE}^n(y_n, \hat{y}_n) = -y_n \log \hat{y}_n, \tag{3}$$

where $y_n$ represents the one-hot encoded vector of the ground-truth label, $\hat{y}_n$ is the predicted logits (i.e., the softmax output of the network), and $\lambda_o$ is a loss balance weight. A higher $\lambda_o$ will prioritize the old task performance over the new task. During training, a new batch gets passed through the old network to record its outputs, which then are used in $\mathcal{L}_{KD}$ to promote that the network updates do not deviate from the old network.

**Incremental Classifier and Representation Learning** (iCaRL) (Rebuffi et al., 2017) is an early attempt at rehearsal approaches for class-incremental learning. It leverages memory replay and KD regularization. After each task's training, it utilizes the herding sampling technique (Welling, 2009) to select a small set of exemplars (i.e., representative samples) from the current task's training data and stores them in memory. Herding works by choosing samples that are closest to the centroid of each old class. When the next task becomes available, the in-memory exemplars will be combined with the new task's training data to update the network. The loss function of iCaRL is the same as Equation 1 in LwF, which is a combination of cross-entropy (CE) loss on new classes and the KD loss on old classes to allow knowledge transfer. Additionally, Rebuffi et al. (2017) added a nearest-mean-of-exemplars (NME) classifier which performs a forward pass through the model for the holdout data and classifies new samples by finding the the nearest class mean.

**Bias Correction** (BiC) (Wu et al., 2019) introduces a BiC correction layer after the last FC layer to adjust the weights in order to tackle the imbalance problem. There are two stages of training. In the first stage, the network will be trained with new task's data and in-memory samples of old tasks using the CE and KD loss similar to iCaRL. In the second stage, the layers of the network are frozen and a linear BiC layer is added to the end of the network and trained with a small validation set consisting of samples from both old and new tasks. The linear model of the BiC layer corrects the bias on the output logits for the new classes:

$$\tilde{y}_k = \begin{cases} \hat{y}_k & 1 \le i \le n \\ \alpha \hat{y}_k + \beta, & n+1 \le i \le n+m \end{cases}$$

where $\hat{y}_k$ is the output logits on the $k$th class, the old classes are $[1, ..., n]$ and the new classes are $[n+1, ..., n+m]$. $\alpha$ and $\beta$ are the bias parameters of the linear model, which are optimized in the following loss function:

$$L_{BiC} = -\sum_{k=1}^{n+m} \delta_{y=k} \log \mathtt{softmax}(\tilde{y}_k),$$

where $\delta_{y=k}$ is the indicator function to check if the ground-truth label $y$ is the same as a class label $k$. The intuition is that a balanced small validation set for all seen classes can counter the accumulated bias during the training of the new task. As the non-linearities in the data are already captured by the model's parameters, the linear function proves to be quite effective in estimating the bias in the output logits of the model arising.

## A.3    HAR Datasets

*Physical Activity Monitoring* (PAMAP2) (Reiss & Stricker, 2012) contains 18 physical activities, including a wide range of everyday, household and sport activities, from 9 users. The sensor data is collected on accelerometers worn on the chest, dominant arm and side ankle, totaling 10 hours of data with a sampling frequency of 100Hz. We use features generated by Wang et al. (2018a) which contain 12 classes and 6 users. Their feature generation protocol was that 27 features are extracted per sensor on each body part which results in 81 per body part, including mean, standard deviation, and spectrum peak position. *Daily and Sports Activities Dataset* (DSADS) (Altun et al., 2010) contains 19 activity classes such as running, rowing, and sitting. Each user performs each of these activities for 5 minutes at a 25Hz sampling frequency. The data samples are collected on 8 users with 5 accelerometer units on each user's torso and extremities. The subjects were not instructed on how the activities should be performed. Here we use the same set of feature generation protocol as PAMAP2 (Wang et al., 2018a). *Human Activity Recognition Dataset* (HAPT) (Reyes-Ortiz et al., 2016) contains 12 activities such as standing or walking and postural transitions such as stand-to-sit. It is collected from 30 subjects wearing a smartphone (Samsung Galaxy S II) on their waist with a 50Hz sampling frequency. The data is split with a fixed-width sliding window of 2.56 seconds and 50% overlapping. After filters and transformations, features like mean, absolute difference, skewness and more are generated. In the end, Reyes-Ortiz et al. Reyes-Ortiz et al. (2016) extracted 561 features from accelerometer and gyroscope sensor readings. To ensure uniform evaluation across all users and classes, we apply a filter to each dataset. This involves removing users who do not have data for all classes and eliminating classes that are significantly underrepresented. As a result 1 class and 3 users respective 2 users are removed from PAMAP2 and HAPT.

## A.4    Without Hadamard transformation

R4: If we do not apply the Hadamard transformation in the backward pass, out performance drops to an averaged accuracy of 48.11 with a standard deviation of 21.4 over 5 runs for iCaRL on HAPT. One can see that without the Hadamard transform the performance is significantly compromised in terms of accuracy and stability.

## A.5    Hyperparameters

For the CIFAR datasets we used the hyperparameters reported in Zhou et al. (2023), and did not employ hyperparameter search techniques for any reported results. We use 0.05 as learning rate for CIFAR100 and 0.01 for the HAR datasets. We apply a decay of 0.1 at the 60th, 100th and 140th epoch for CIFAR100 and at the 50th for HAR with a weight decay of 0.0002 for both. The batch size is 128. For techniques that employ memory replay, CIFAR100 memory size is set to 2000 and HAR's too 200 as the datasets are smaller. The number of samples per class are reduced at every step to make space for new classes. CIFAR100 models are trained for 170 epochs per task and 100 for HAR. Both use SGD with a momentum of 0.9. The temperature value for KD is set to two as hinted by Hinton et al. (2015) and the lambda multiplier to three. BiC's split ratio is 0.1. CIFAR100 models are introduced to 20 classes per task and HAR to 2.

## A.6    Tile size variations

R1: To determine a good tile-size, we performed a hyperparameter search with iCaRL on HAPT. Table 4 shows that there is a stable performance until tile-size 32. Larger ones start to compromise accuracy performance and increase variance.

## A.7    No Hadamard Requantization

R1: If the Hadamard re-quantization is left out we achieve the average final accuracy of 81.18 with a standard deviation of 3.67 over 5 runs for iCaRL on HAPT. Note that compared to the re-quantized performance in table 2 this difference is not significant.

Table 4: Results for HDQT using different tile sizes. Numbers are final accuracy percentages for iCaRL averaged over 5 runs on the HAPT dataset.

| | | | | Tile-size | | | | |
|---|---|---|---|---|---|---|---|---|
| | 1 | 2 | 4 | 8 | 16 | 32 | 64 | 128 |
| iCaRL | 82.04 | 83.18 | 82.26 | 81.64 | 81.64 | 82.39 | 80.17 | 41.16 |
| | ($\pm$2.62) | ($\pm$2.88) | ($\pm$1.71) | ($\pm$1.07) | ($\pm$3.41) | ($\pm$3.68) | ($\pm$4.24) | ($\pm$37.19) |

## A.8 NO HADAMARD ENERGY

Figure 5 displays energy numbers if no Hadamard transform is used.

Table 5: Accelerator cost-model derived estimates for the energy incurred to compute two representative layers of the FCN for HAPT. We include all operations, *e.g.,* max, FHT, etc. Results indicate an improvement of up to $4.0\times$ over the FP16 baseline architecture. R1: * indicates results for integer quantization without Hadamard transformation.

| Layer | Inpt. BW | Accm. BW | Inpt. Dim. | Wgt. Dim. | FWD Energy $\mu$ J | BWD Energy $\mu$ J | Total Energy $\mu$ J | Improvement (vs FP16) |
|---|---|---|---|---|---|---|---|---|
| Hidden | FP16 | FP16 | $128 \times 561$ | $561 \times 561$ | 1630.6 | 3319.8 | 4950.4 | 1.0 |
| | 4* | 16* | $128 \times 561$ | $561 \times 561$ | 86.2 | 496.1 | 582.3 | 8.5 |
| | 4* | 12* | $128 \times 561$ | $561 \times 561$ | 80.3 | 482.7 | 563.0 | 8.8 |
| | 4* | 8* | $128 \times 561$ | $561 \times 561$ | 73.7 | 149.1 | 222.8 | 22.2 |
| Output | FP16 | FP16 | $128 \times 561$ | $11 \times 561$ | 45.0 | 88.9 | 133.8 | 1.0 |
| | 4* | 16* | $128 \times 561$ | $11 \times 561$ | 4.0 | 8.1 | 12.1 | 11.1 |
| | 4* | 12* | $128 \times 561$ | $11 \times 561$ | 3.9 | 7.8 | 11.7 | 11.4 |
| | 4* | 8* | $128 \times 561$ | $11 \times 561$ | 3.7 | 7.6 | 11.3 | 11.8 |

## A.9 LUQ CODE MODIFICATIONS

We modified the LUQ code (found in the supplementary materials under `https://openreview.net/forum?id=yTbNYYcopd`)to account for: i) difference between `max()` and `max(abs())` which resulted in incorrect scaling from intended dynamic range (see `Luq supplementary/LUQ_code/vision/models/modules/LUQ.py` line 103), ii) LUQ having 17 distinct levels for 4-bit due to overcounting (see `Luq supplementary/LUQ_code/vision/models/modules/LUQ.py` lines 24, 25, 27, and 48 ), and iii) the omission of resetting the positive values upper bound when of negative activations are detected, resulting in 25 distinct representation levels for four-bit quantization (see `Luq supplementary/LUQ_code/vision/models/modules/LUQ.py` line 48). For ii we attempt to retain their symmetry at the cost of one quantization level. Results obtained with the modified version of their code are marked with a * in Table 2.

## A.10  NME

For our experiments, we have removed the NME classifiers from iCaRL because we are interested in the network learning interaction between quantization and CIL solely. Table 6 shows a comparison for iCaRL with NME with no additional cross influence with quantization.

Table 6: Result for standard learning (NoCL) and CIL with 20 classes (CIFAR100) or 2 classes (DSADS, PAMPA2, or HAPT) per task quantized (4-bit inputs and 8-bit accumulators) and unquantized (FP). Numbers are final accuracy percentages averaged over 20 runs of the HAR datasets and 5 runs for CIFAR100. Small numbers in brackets indicate standard deviation. Note that NME gives iCaRL a straight forward performance boost that is not influenced by quantization.

|  | CIFAR100 | | DSADS | | PAMAP2 | | HAPT | | |
|---|---|---|---|---|---|---|---|---|---|
|  | FP | 4-bit | FP | 4-bit | FP | 4-bit | FP | 4-bit | $\mathbb{E}[\Delta\text{Acc.}]$ |
| NoCL | 68.61 | 59.43 | 84.44 | 82.71 | 85.81 | 84.35 | 93.15 | 92.08 | *3.36* |
|  | ($\pm$0.24) | ($\pm$1.88) | ($\pm$4.36) | ($\pm$4.45) | ($\pm$4.91) | ($\pm$7.66) | ($\pm$2.11) | ($\pm$2.10) | |
| iCaRL NME | 52.84 | 47.15 | 76.38 | 77.19 | 80.03 | 79.96 | 85.80 | 84.94 | *1.45* |
|  | ($\pm$0.74) | ($\pm$1.75) | ($\pm$4.06) | ($\pm$4.22) | ($\pm$5.57) | ($\pm$4.53) | ($\pm$2.93) | ($\pm$3.24) | |
| iCaRL | 43.20 | 37.25 | 72.83 | 72.74 | 77.52 | 77.87 | 82.98 | 82.56 | *1.53* |
|  | ($\pm$1.21) | ($\pm$2.42) | ($\pm$4.84) | ($\pm$4.59) | ($\pm$5.50) | ($\pm$5.02) | ($\pm$5.12) | ($\pm$4.51) | |

## A.11  COMPREHENSIVE ENERGY ESTIMATION

Table 7: Accelerator cost-model derived estimates for the energy-cost incurred for implementing two representative layers of the FCN for DSADS, PAMAP2, and HAPT. These include all operations required, such as `max` or `FHT`. **H** stands for hidden layer and **O** for output layer.

| Layer | Inpt. BW | Accm. BW | Inpt. Dim. | Wgt. Dim. | FWD Energy $\mu$J | BWD Energy $\mu$J | Total Energy $\mu$J | Improvement (vs FP16) |
|---|---|---|---|---|---|---|---|---|
| DSADS **H** | FP | FP | $128 \times 405$ | $405 \times 405$ | 905.3 | 1840.0 | 2745.3 | 1.0 |
|  | 4 | 8 | $128 \times 405$ | $405 \times 405$ | 46.7 | 803.6 | 850.3 | 3.2 |
| DSADS **O** | FP | FP | $128 \times 405$ | $405 \times 19$ | 49.5 | 99.6 | 149.1 | 1.0 |
|  | 4 | 8 | $128 \times 405$ | $405 \times 19$ | 5.6 | 23.2 | 28.8 | 5.1 |
| PAMAP2 **H** | FP | FP | $128 \times 243$ | $243 \times 243$ | 585.4 | 904.0 | 1489.4 | 1.0 |
|  | 4 | 8 | $128 \times 243$ | $243 \times 243$ | 17.0 | 93.4 | 110.4 | 13.5 |
| PAMAP2 **O** | FP | FP | $128 \times 243$ | $243 \times 11$ | 20.1 | 37.7 | 57.8 | 1.0 |
|  | 4 | 8 | $128 \times 243$ | $243 \times 11$ | 2.9 | 12.9 | 15.8 | 3.7 |
| HAPT **H** | FP16 | FP16 | $128 \times 561$ | $561 \times 561$ | 1630.6 | 3319.8 | 4950.4 | 1.0 |
|  | 4 | 8 | $128 \times 561$ | $561 \times 561$ | 87.0 | 991.3 | 1078.3 | 4.5 |
| HAPT **O** | FP16 | FP16 | $128 \times 561$ | $561 \times 11$ | 45.0 | 88.9 | 133.8 | 1.0 |
|  | 4 | 8 | $128 \times 561$ | $561 \times 11$ | 6.4 | 28.7 | 35.1 | 3.8 |

A.12    ADDITIONAL FIGURES: EFFECTS OF FORGETTING FROM QUANTIZATION AND CIL

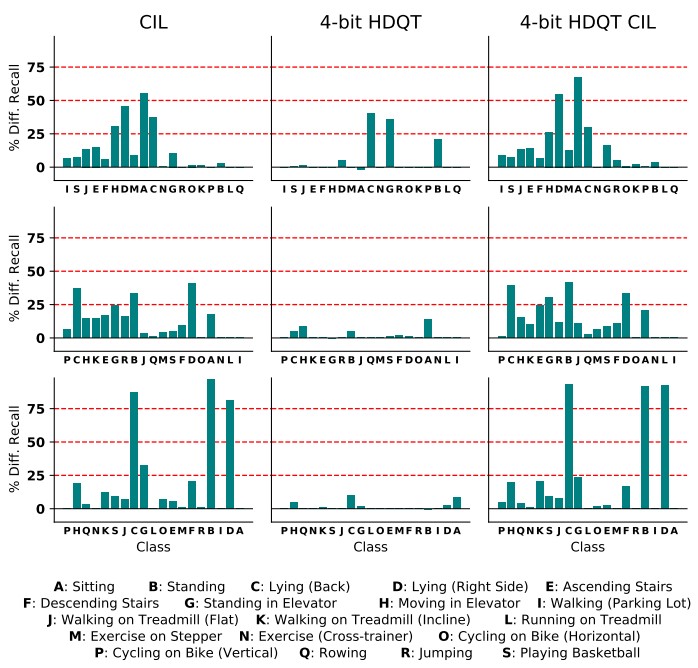

Figure 5: Showing class accuracy difference comparison of an unquantized CIL model with models trained with only HDQT and one with CIL and HDQT. Training was performed on all 19 classes of the DSADS dataset (more examples in A.12). Classes are sorted in incoming CIL order. Note that quantization does not add or amplify CF *e.g.,* class accuracy for H (Moving in Elevator) increases while A (Sitting) goes down when combined. Additionally, an overview of per-class forgetting due to CIL and quantization for three different random seeds on DSADS.

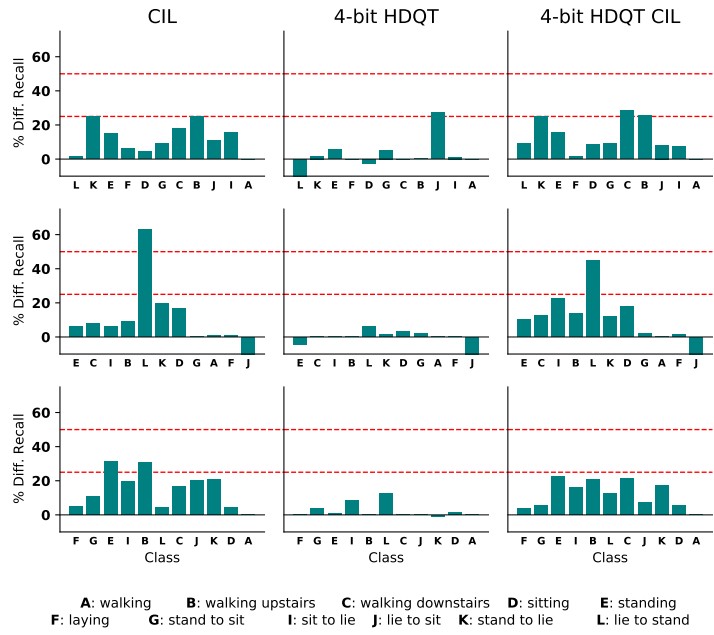

Figure 6:   Additional overview of per-class forgetting due to CIL and quantization for three different random seeds on HAPT.

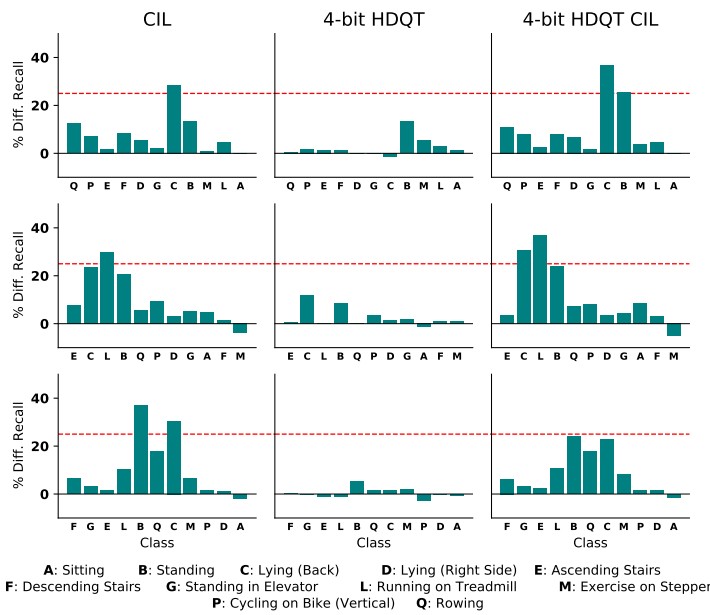

Figure 7: Additional overview of per class forgetting due to CIL and quantization for three different random seeds on PAMAP2.

### A.13 GRADIENT BIAS AND STANDARD DEVIATION W/WO STOCHASTIC ROUNDING

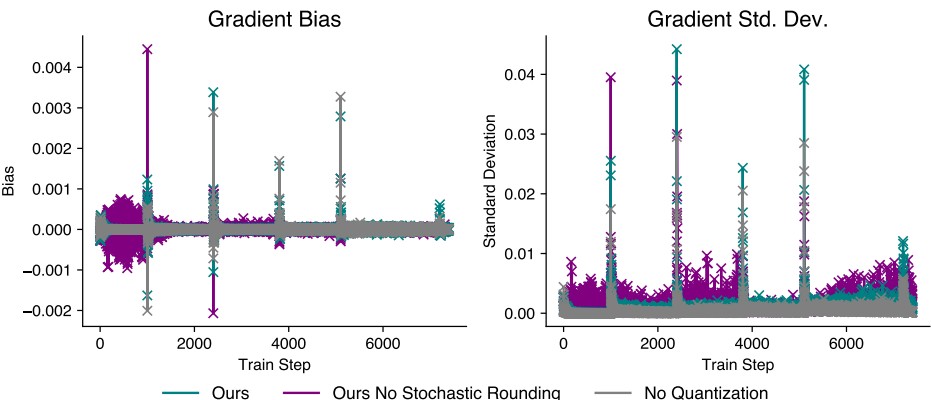

Figure 8: R1: Showing the development of the gradient bias and standard deviation over CL training. Note how especially at the beginning the bias when not using stochastic rounding is different from an unquantized baseline which is remedied through stochastic rounding. Additionally, stochastic rounding keeps the standard deviation smaller.

### A.14 GRADIENT DISTRIBUTIONS OF CL NOQ AND FQT

R1: Figures 9 and 10 show the distributions over the gradients of the first batch for every task. One can see that at lower bit-widths HDQT is more true to the unquantized distributions. We think this is the reason for the better performance at lower bit-widths. For 16-bits accumulators this obvious difference is not seen and both methods are able to learn.

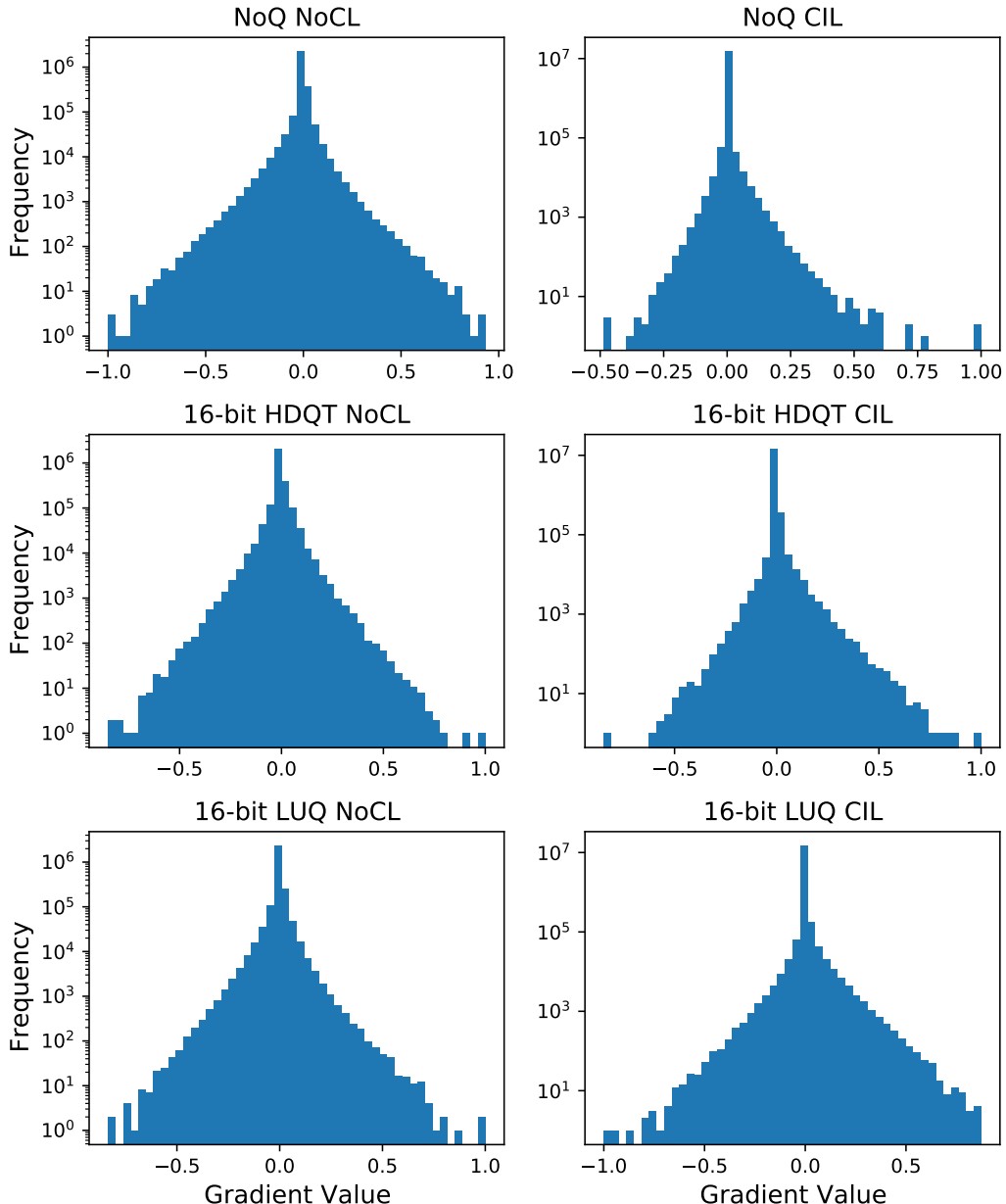

Figure 9: R1: Gradient distributions of standard learning vs continual learning for HDQT and LUQ at 4-bits input and 16-bits accumulator quantization.

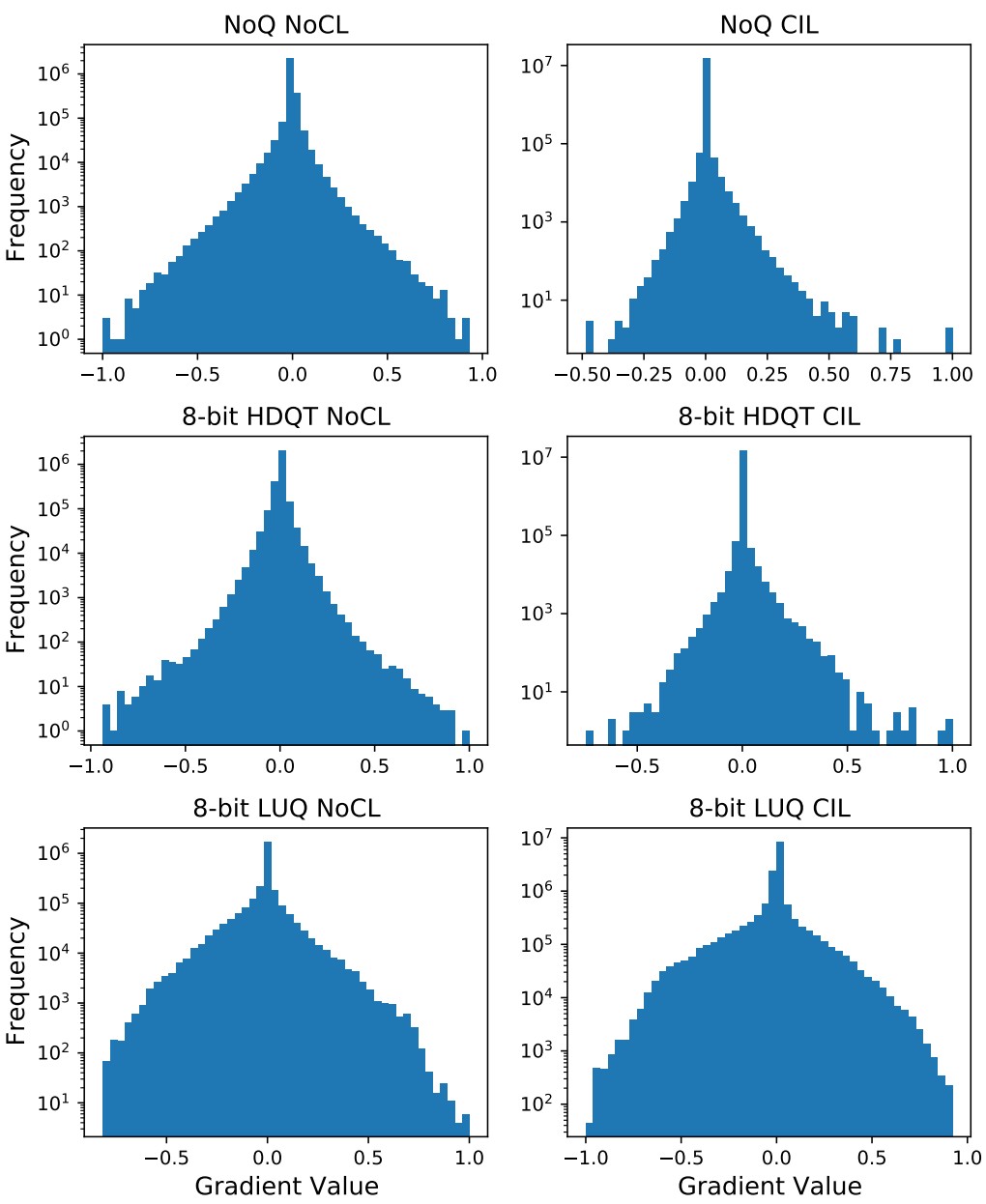

Figure 10: R1: Gradient distributions of standard learning vs continual learning for HDQT and LUQ at 4-bits input and 8-bits accumulator quantization.

