# OpenReview forum: "Hadamard Domain Training with Integers for Class  Incremental Quantized Learning"
_ICLR.cc/2024/Conference — Submitted to ICLR 2024_

### Official Review · Reviewer_2d5P · 2023-10-29

**Soundness:** 2 fair
**Presentation:** 1 poor
**Contribution:** 2 fair
**Rating:** 5
**Confidence:** 4

**Summary:**

This paper try to deal with two important subjects in deep learning field at the same time. - Quantization and Continual learning.
Quantization is a promising way to reduce costs of deep neural networks inference. Furthermore, there is a way to reduce training cost that is much more huge than inference cost by using low bit numerical formats, which is called FQT (Fully Quantized Training). CL (Continual learning) is a research field that try to handle a realistic scenario; the number of categories are not fixed and new categories and data can be added while previous ones aren’t available.

However, FQT and training under CL scenarios are both harmful to the performance of networks. Thus, it is obvious that executing FQT under CL scenarios leads to unacceptable performance degradation. That’s why it is hard to find a trial to deal with both subjects simultaneously, even though these two topics are much important in practice.

To solve this problem, the paper introduces Hadamard transform, which is an inexpensive operation, in the training process. By applying Hadamard transform and stochastic rounding to the backward pass of training process, the proposed method make FQT possible under CL scenarios, with little to no sacrifice in performance.

**Strengths:**

- This paper is the very first work that trying to solve FQT and CL simultaneously.
- There is little performance degradation, despite that FQT is executed under continual learning scenarios.

**Weaknesses:**

- Previous CL works that the paper cites were presented years ago, and the paper doesn’t address recent works.
- The paper doesn’t show experimental results that use larger and more realistic datasets, such as ImageNet.
- The formula of Hadamard transform is wrong; $H_k = [H_k  \quad H_{k-1};   H_{k-1} \quad  -H_{k-1}]$ -> $H_k = \frac{1}{\sqrt{2}} [H_k    \quad H_{k-1};   H_{k-1} \quad -H_{k-1}]$
- There are some figures that are hard to recognize.
    - In Figure 2, the first three rows compare standard quantization and Hadamard quantization, and the last row is an ablation study of quantizing tensors in the backward pass and accumulator quantization. However, because they aren’t properly distinguished, it seems that the last row also shows a comparison between standard quantization and Hadamard quantization.
    - Likewise, in Figure 4, the first three rows and the last row should have been distinguished.
- In Table 2, a 4-bit input and 12-bit accumulator for previous works is critical to compare the proposed method with other works. However, those results aren’t provided.

**Questions:**

- There already is a previous work that applies the Hadamard transform to quantization. What’s different between the previous work and the proposed method?
- It seems too harsh to apply FQT and CL simultaneously. Can an example of a scenario that should apply FQT and CL together be provided?
- It helps understand how effective the proposed method is if results about FQT under CL scenarios without Hadamard transform are provided. Can the results be provided?

---

> ### Author Response · Authors · 2023-11-21
>
> Thanks for your clear and constructive comments. We sincerely appreciate your time in reading the paper. Our point-to-point responses to your comments are given below.  In the paper, you can identify changes attributed to your reviews by R4.
>
> R4: Previous CL works that the paper cites were presented years ago, and the paper doesn’t address recent works.
>
> Authors: We chose to address fundamental continual learning techniques that are exemplary for different research directions in continual learning. More recent techniques are often at least partly reformulations of ideas presented in the investigated techniques. Therefore, these techniques are usually part of the comparison techniques in continual learning e.g. in [1] and we argue that they provide a good indication of the connectivity of CL and FQT.
>
> [1] Zhou, Da-Wei, et al. "Deep class-incremental learning: A survey." arXiv preprint arXiv:2302.03648 (2023).
>
> R4: The paper doesn’t show experimental results that use larger and more realistic datasets, such as ImageNet.
>
> Authors: The target for our technique are limited computational resources scenarios for example on edge devices detecting human activity or AR/VR scenarios. Hence, the datasets we are most interested in are relatively small. We included CIFAR100 with a ResNet architecture to allow the continual learning community to compare results to previous work in the field. Quantization is usually resulting in performance degradation as information is lost in the process. In comparison with the other FQT techniques though, we achieve competitive results on the relatively large CIFAR100.
>
> R4: The formula of Hadamard transform is wrong;
>
> Authors: While the normalization factor is often omitted (note that we had mentioned the transform is orthogonal and not orthonormal) we have now included this normalization factor for clarity.
>
> R4: There are some figures that are hard to recognize. In Figure 2, the first three rows compare standard quantization and Hadamard quantization, and the last row is an ablation study of quantizing tensors in the backward pass and accumulator quantization. However, because they aren’t properly distinguished, it seems that the last row also shows a comparison between standard quantization and Hadamard quantization. Likewise, in Figure 4, the first three rows and the last row should have been distinguished.
>
> Authors: We have added different coloured backgrounds to the mentioned figures 2 and 4 and now address them accordingly in the paper.
>
> R4: In Table 2, a 4-bit input and 12-bit accumulator for previous works is critical to compare the proposed method with other works. However, those results aren’t provided.
>
> Authors: We have added results for 12 bits LUQ in Table 2. FP134 already fails for 16 bits and also fails for 12. Therefore, our HDQT method is the only one that does not fail with 12 bits or lower for the accumulator.

---

> ### Author Response · Authors · 2023-11-21
>
> Questions:
>
> R4: There already is a previous work that applies the Hadamard transform to quantization. What’s different between the previous work and the proposed method?
>
> Authors: While previous work [1] (which was cited as (Xi et al., 2023) in our paper) has indeed used the Hadamard transform during the forward pass of a model ( to account for outliers in the forward pass activations), this is not our approach. Instead, we apply the Hadamard transform to the backward pass to better spread information across multiple quantizer levels and capture variety in gradient distributions and learning dynamics of CL. We furthermore identify pain points such as accumulator bit-width in the forward pass (note that it is not necessary in the backward pass) and demonstrate techniques to mitigate any accuracy loss owing to quantization.
>
> R4: It seems too harsh to apply FQT and CL simultaneously. Can an example of a scenario that should apply FQT and CL together be provided?
>
> Authors: We are not the first that are interested in applying continual learning to the edge; e.g. [1], [2], [3] or this late breaking paper [5]. Continual learning with the ability to learn new tasks over time without the need to train from scratch has been gaining more and more attention in applications with resource-constrained devices; for example, in AR/VR devices where instantaneous adaptation might be needed, or small drones or mobile robots which need to adapt to new environments in real-time while operating with limited memory and computational power. There have also been examples of using continual recalibration on brain-computer interface devices [4] where miscalibration might result in the non-detection of seizures. Quantization leverages the limited computational resources and enables continual learning where it would otherwise be impossible either due to practical or legal reasons.  Or alternatively, enables larger models on the same resource-constrained device. Additionally, note that energy and time typically do not trade off and we are reporting energy estimates rather than power estimates.
>
>
> [1] Wang, Zifeng, et al. "SparCL: Sparse continual learning on the edge." Advances in Neural Information Processing Systems 35 (2022): 20366-20380.
>
> [2] Hayes, Tyler L., and Christopher Kanan. "Online continual learning for embedded devices." arXiv preprint arXiv:2203.10681 (2022).
>
> [3] Pietroń, Marcin, et al. "Ada-QPacknet--adaptive pruning with bit width reduction as an efficient continual learning method without forgetting." arXiv preprint arXiv:2308.07939 (2023).
>
> [4] Fan, Chaofei, et al. "Plug-and-Play Stability for Intracortical Brain-Computer Interfaces: A One-Year Demonstration of Seamless Brain-to-Text Communication." arXiv preprint arXiv:2311.03611 (2023).
>
> [5] Verwimp et al. "Continual Learning: Applications and the Road Forward" arXiv preprint arXiv:2311.11908 (2023)
>
>
> R4: It helps understand how effective the proposed method is if results about FQT under CL scenarios without Hadamard transform are provided. Can the results be provided?
>
> Authors: We have added results for FQT without the Hadamard transform in Appendix A4. We see a significant performance reduction without it e.g. in the case of iCaRL on HAPT the average accuracy drops from 82.56 to 48.11 and the standard deviation increases from 4.51 to 21.4. Therefore, without the Hadamard transform FQT is severely compromised in terms of accuracy and stability.

---

### Official Review · Reviewer_q1uv · 2023-10-30

**Soundness:** 2 fair
**Presentation:** 2 fair
**Contribution:** 2 fair
**Rating:** 5
**Confidence:** 2

**Summary:**

This submission proposed to improve Fully Quantized Training (FQ) by introducing an extrac matrix H and its corresponding inverse H^{-1} into matrix multiplication W^\top X. The introduced H provide re-allocation for W to enhance usage of quantization bits.

**Strengths:**

1. It is interesting to borrow method from other fields (Hadamard Domain) to Quantization-aware Training (QAT).

**Weaknesses:**

1. The introudction of Hadamard Domain is missing. Readers unfamiliar with the field are confused with its application in this submission.
2. Introducing extrac matrix H brings in more parameters for training. Besides, how to determine the parameter H (trainable or pre-calculated) is not mentioned.
3. The proposed algorithm is not connected to Class Incremental Learning (CIL).

**Questions:**

I don't have question currently. Please clarify weakness 2&3.

**Details Of Ethics Concerns:**

N.A.

---

> ### Author Response · Authors · 2023-11-21
>
> Thank you so much for taking the time to review our academic paper. Your thoughtful feedback is truly invaluable, and we appreciate the constructive insights you've provided.  In the paper, you can identify changes attributed to your reviews by R3.
>
> R3: The introudction of Hadamard Domain is missing. Readers unfamiliar with the field are confused with its application in this submission.
>
> Authors: We appreciate the reviewer’s perspective and had originally omitted a detailed introduction of the Hadamard transform due to the space limitations of the paper. We have added additional text and references to provide additional information for interested readers to address this concern.
>
> R3: Introducing extra matrix H brings in more parameters for training. Besides, how to determine the parameter H (trainable or pre-calculated) is not mentioned.
>
> Authors: H is not a trainable parameter but a predefined block diagonal matrix (consisting of +1, -1, and 0 entries), with the size of the matrix determined by the dimensionality of the tensor to be quantized. If we understand this concern correctly, the question is how we determine the composition of the H matrix. H is a block diagonal matrix composed of several sylvestre matrices which we determine the individual sizes by choosing the minimum amount of individual sylvestre matrices that fit. This is done only once for hidden layers and for output layers every time new classes are encountered which changes the dimensionality of the output.
>
> R3: The proposed algorithm is not connected to Class Incremental Learning (CIL).
>
> We respectfully disagree and would like to highlight that our technique is reciprocally advantageous together with continual learning for the following reasons:
> 1) In Figure 2, we show that the gradient distributions encountered during the CL learning process are not favourable for uniform integer quantization. We empirically demonstrate that this is mitigated through the use of the spreading properties of the Hadamard transform.
> 2) We use an adaptive batch-dependent quantization range to deal with the changing data distributions from CL.
> 3) Our simplicity is an advantage for CL because we have no need for data samples or other parameter storage. This allows us to preserve resources for the often intensive CL techniques[1,2].
>
> [1] Hayes, Tyler L., and Christopher Kanan. "Online continual learning for embedded devices." arXiv preprint arXiv:2203.10681 (2022).
>
> [2] Verwimp et al. "Continual Learning: Applications and the Road Forward" arXiv preprint arXiv:2311.11908 (2023)

---

> > ### Comment · Reviewer_q1uv · 2023-11-23
> > **Response to Author**
> >
> > Thanks for your detailed response.
> >
> > About Weakness 2 (Introduction of matrix): I am aware of how the H is calculated. Thanks for your clarification. Author should add more details on the calculation in the context.
> >
> > About  Weakness 3 (Connection with CIL):
> > 1. "adaptive batch-dependent quantization range" is not related to the Hadamard transform.
> > 2. Spreading properties of the Hadamard transform is not clearly stated and I am not familiar with it. But it seems to be a very nice properties in CL. Please add more explanation on that.
> > 3. The advantage of the method should be related to Hadamard transform. If so, please provide more explanation on the reason and its usage.
> >
> > As I am still very unfamiliar with Hadamard transform, I maintain my score and look forward to more explanation on that.

---

### Official Review · Reviewer_YCXn · 2023-10-31

**Soundness:** 2 fair
**Presentation:** 2 fair
**Contribution:** 3 good
**Rating:** 5
**Confidence:** 3

**Summary:**

This paper introduces a technique using Hadamard transforms for low-precision training with integer matrix multiplications, addressing the accuracy degradation issue of aggressive quantization. Also, the authors further determine which tensors need stochastic rounding and propose tiled matrix multiplication to enable low-bit width accumulators. With this, the proposed method, proven on human activity recognition datasets and CIFAR100, achieves accuracy degradation of less than 0.5% and 3% and quantizes all matrix multiplication inputs to 4 bits using an 8-bit accumulator.

**Strengths:**

* This work is clearly written and proposes a principled and efficient method of solving the non-uniformity of gradients during backward propagation.
* Whereas previous works have proposed log scaling of values, this is impractical to implement efficiently on hardware. Also, it is difficult to guess the necessary value range beforehand. The Hadamard transform is both efficient to implement and is ideally suited to removing gradient spikes while preserving information. As such, there is potential for Hadamard domain backpropagation in reduced-precision training.
* In the backpropagation process, utilizing HDQT allows setting the input and accumulation bit-width to 4 and 8 bits, respectively, showcasing improved energy efficiency compared to conventional CIL methods.
* The authors present promising results on image datasets (CIFAR100) and human activity recognition datasets (DSADS, PAMAP2, HAPT), demonstrating a degradation of ≤ 2.5% through quantization in vision tasks.

**Weaknesses:**

* It is difficult to understand the motivation for connecting continual learning with integer valued training. While edge devices have fewer resources for training, they would not be expected to perform much training or have it completed quickly. Therefore, it would be difficult to justify the additional expense in both software and hardware when the same result can be obtained simply by waiting for a few more hours.
* More experiments on different domains are necessary to support the claims of the efficacy of Hadamard domain quantization. For a fairer comparison, there should be a comparison using other methods used in fully quantized training when applied to continual learning. Moreover, there should be a comparison with previous methods when training from scratch, not just during continual learning.
* The method proposed in the paper would be difficult to implement due to the requirement for stochastic rounding, which cannot be easily implemented in hardware, integer-based or otherwise. Even if such hardware could be implemented, this would require custom circuitry solely for that purpose.

**Questions:**

* In Section 4.4, when discussing the energy estimates derived from the accelerator cost model for HDQT application, there is a concern that if the training time increases due to HDQT, it may consume more energy than conventional methods. Can the experiments include a comparison of training times when applying HDQT?
* When examining the results for CIFAR100 in Table 1, the accuracy degradation appears to be relatively higher compared to other datasets. It seems that the trend may vary depending on the dataset or network architecture. Have there been results applying the method to more complex network architectures or larger datasets?
* While there are advantages to performing matrix multiplication as integer-arithmetic units through HDQT, it is likely that the overhead for Quantization and Stochastic Quantization cannot be ignored, especially as the batch size increases. I am curious about the energy efficiency, including the overhead for Quantization and Stochastic Quantization, as the batch size becomes larger.

---

> ### Author Response · Authors · 2023-11-21
>
> Thank you for your constructive critique and suggestions, they are very helpful for us to improve our paper. We have carefully integrated them into the revised paper. In the following, your comments are first stated and then followed by individual responses.  In the paper, you can identify changes attributed to your reviews by R2.
>
> R2: It is difficult to understand the motivation for connecting continual learning with integer-valued training. While edge devices have fewer resources for training, they would not be expected to perform much training or have it completed quickly. Therefore, it would be difficult to justify the additional expense in both software and hardware when the same result can be obtained simply by waiting for a few more hours.
>
> Authors: We are not the first that are interested in applying continual learning to the edge; e.g. [1], [2], [3] or this late breaking paper [5]. Continual learning with the ability to learn new tasks over time without the need to train from scratch has been gaining more and more attention in applications with resource-constrained devices; for example, small drones or mobile robotics need to adapt to new environments fast in real-time while with limited memory and computational power. Or the recent example of continual recalibration on brain-computer interface devices [4] also opens opportunities. Quantization leverages the limited computational resources and enables continual learning where it would otherwise be impossible either due to practical or legal reasons. Or alternatively, enables larger models on the same resource-constrained device. Additionally, note that energy and time typically do not trade off and we are reporting energy estimates rather than power estimates.
>
>
> [1] Wang, Zifeng, et al. "SparCL: Sparse continual learning on the edge." Advances in Neural Information Processing Systems 35 (2022): 20366-20380.
>
> [2] Hayes, Tyler L., and Christopher Kanan. "Online continual learning for embedded devices." arXiv preprint arXiv:2203.10681 (2022).
>
> [3] Pietroń, Marcin, et al. "Ada-QPacknet--adaptive pruning with bit width reduction as an efficient continual learning method without forgetting." arXiv preprint arXiv:2308.07939 (2023).
>
> [4] Fan, Chaofei, et al. "Plug-and-Play Stability for Intracortical Brain-Computer Interfaces: A One-Year Demonstration of Seamless Brain-to-Text Communication." arXiv preprint arXiv:2311.03611 (2023).
>
> [5] Verwimp et al. "Continual Learning: Applications and the Road Forward" arXiv preprint arXiv:2311.11908 (2023)
>
>
> R2: More experiments on different domains are necessary to support the claims of the efficacy of Hadamard domain quantization. For a fairer comparison, there should be a comparison using other methods used in fully quantized training when applied to continual learning. Moreover, there should be a comparison with previous methods when training from scratch, not just during continual learning.
>
> Authors: We have shown the efficacy of our method on three datasets recorded using different devices including accelerometers and gyroscopes in the human activity recognition domain, which are representative tasks for energy-constraint edge domains where adaptation is necessary (e.g. AR/VR). Additionally, we provide results on CIFAR100 for a better comparison to related work in CL. While more datasets can definitely strengthen our case, we believe that for our target use case we have shown sufficient datasets and techniques to demonstrate the efficacy of our method.  Table 2 shows the performance of two state-of-the-art FQT methods LUQ and FP134  applied to the CIL scenario. Note that those do not use integer quantization and instead resort to custom floating point formats.  To further address your concern, we have added NoCL results for LUQ and FP134 in Table 2, which show that they are incapable of learning with the same our low bit-width setting.
>
> R2: The method proposed in the paper would be difficult to implement due to the requirement for stochastic rounding, which cannot be easily implemented in hardware, integer-based or otherwise. Even if such hardware could be implemented, this would require custom circuitry solely for that purpose.
>
> Authors: Stochastic rounding is used in other FQT e.g. [1,2] and we are possibly facing a chicken-and-egg problem where hardware designers won’t put support in accelerators due to its being unproven and software designers can’t use it due to no hardware support. We believe, however, that our method shows promise and might over time garner similar widespread appeal similar to how earlier papers had fp8 and int4 software demonstrations which were only after multiple efforts supported natively within accelerators.

---

> ### Author Response · Authors · 2023-11-21
>
> Questions:
>
> R2: In Section 4.4, when discussing the energy estimates derived from the accelerator cost model for HDQT application, there is a concern that if the training time increases due to HDQT, it may consume more energy than conventional methods. Can the experiments include a comparison of training times when applying HDQT?
>
> Authors: There are two ways in which the training time can increase: i) more training steps to reach acceptable accuracy, and ii) latency increase due to hardware implementations. For i) we would like to highlight that all our training runs are all run for the same number of epochs and the energy number we present are for a single step, e.g. the gains scale with training length which however is the same for quantized and unquantized runs. For ii) typically, when power and time are traded off, this is done by increasing the voltage/clock of the hardware which in turn reduces delay (at the cost of increased power). Note, however, that we report energy which is power x delay which is approximately invariant to this.
>
> R2: When examining the results for CIFAR100 in Table 1, the accuracy degradation appears to be relatively higher compared to other datasets. It seems that the trend may vary depending on the dataset or network architecture. Have there been results applying the method to more complex network architectures or larger datasets?
>
> Authors: The target for our technique are edge devices with limited computational resources in scenarios such as human activity recognition. Hence, the datasets we are most interested in are relatively small. We included CIFAR100 with a ResNet architecture to allow the continual learning community to compare results to previous work in the field. Quantization usually results in performance degradation as information is lost in the process. In comparison with the other FQT techniques though, we achieve competitive results on the relatively large CIFAR100 specifically when fewer bits are used for accumulators where the other FQT methods, despite our best efforts, are unable to deliver results.
>
> R2: While there are advantages to performing matrix multiplication as integer-arithmetic units through HDQT, it is likely that the overhead for Quantization and Stochastic Quantization cannot be ignored, especially as the batch size increases. I am curious about the energy efficiency, including the overhead for Quantization and Stochastic Quantization, as the batch size becomes larger.
>
> Authors: Since our problem domain focused on resource-constrained edge devices, under a class incremental learning setting, we did not study how this would be applied to large batch training. In our studies, the optimal data movement/data flow changes with different batch sizes due to changes in data locality and reuse characteristics.  Disentangling these different hardware impacts while maintaining the focus of this paper on HDQT applied to class incremental learning might not be possible in the space we have available. However, we do provide some answers to the reviewer’s questions after making some simplifying assumptions:
>
> 1) Under a fixed weight-stationary dataflow, then the overhead of implementing the Hadamard transform on the weights is amortized over larger and larger amounts of computation. Because, the Hadamard transform is of the order of N log(N) additions which can be implemented very efficiently, especially in a batched fashion since memory access gets amortized over multiple operations.
> 2) Requantization does not incur any hardware overhead and can be achieved by simply dropping the LSB (least significant bit) during computation. This should not impact the scalability of our approach.
> 3) We note that stochastic quantization has been employed for at large-scale training of e.g. imagenet-scale models [1]. Indeed, stochastic rounding is also included by Qualcomm in their efficient deployment framework: https://quic.github.io/aimet-pages/AimetDocs/api_docs/torch_quantsim.html . It is true that stochastic rounding might require custom hardware to be implemented. Here, we make the case for the development and inclusion of such operations in hardware by demonstrating accuracy improvements.
>
> [1] Wang, Naigang, et al. "Training deep neural networks with 8-bit floating point numbers." Advances in neural information processing systems 31 (2018).
> [2] Chmiel, Brian, et al. "Logarithmic unbiased quantization: Practical 4-bit training in deep learning." (2021).

---

> ### Comment · Reviewer_YCXn · 2023-11-22
>
> Thank you for the detailed answers.
>
> The provided results and details satisfy my concerns. Some of my concerns have been addressed, and my score has improved by one level.

---

> > ### Author Response · Authors · 2023-11-22
> >
> > We appreciate your willingness to update your score. Let us know anything we haven't addressed in sufficient detail and we'll do our best to address them.

---

### Official Review · Reviewer_ZGvQ · 2023-11-04

**Soundness:** 2 fair
**Presentation:** 2 fair
**Contribution:** 1 poor
**Rating:** 3
**Confidence:** 4

**Summary:**

This paper applies fully-quantized training (FQT) to class incremental learning (CIL) by representing activation, weight, and gradient using reduced precision. The paper employs the Hadamard transform for more efficient bit utilization during quantization, applies stochastic rounding to quantization-sensitive activation and gradient in the backward pass, and utilizes tiled quantization for minimal error during partial sum accumulation. The proposed approach enables efficient CIL with 4-bit matrix multiplication and 8-bit accumulation, minimizing performance degradation.

**Strengths:**

- This paper provides a comprehensive background and overview of the proposed approach.
- This paper is the first to apply low-performance degradation FQT to CIL, and it presents results through extensive comparisons with various CIL techniques and existing FQT methods.
- The paper demonstrates the hardware advantages of the proposed 4-bit integer FQT over 16-bit operations through an energy table.

**Weaknesses:**

- This paper lacks novelty. While it introduces FQT to CIL, it is unclear whether the proposed method offers something significantly new compared to the existing FQT with Hadamard transform [1]. Also, the explanation regarding one of the paper's contributions, tiled matrix multiplication, does not provide sufficient details such as hyperparameters like tile size, which are essential for both accuracy and hardware cost. Additionally, the paper does not explain any distinctive aspects compared to chunk-based accumulation proposed in other existing FQT methods [2].
- There is a lack of detailed analysis regarding the overhead incurred by Hadamard transformations in the training process. Furthermore, in [1], Hadamard transformation was used effectively to manage outliers occurring in specific channels of the Transformer model. However, this paper lacks a clear explanation of why the Hadamard transformation is particularly necessary for CIL. In Section 3, the direct relevance of being 'more efficient in using available quantization levels' to performance is not well-established.
- The rationale behind using stochastic rounding is unclear. In Figure 2 (right), the sensitivity of activation and gradient quantization in the backward pass to unbiased quantizers lacks a clear bridge of explanation. There is a lack of description regarding which factors among bias and variance make gradient and activation quantization in the backward pass challenging.
- The hardware comparison is unfair. While this paper emphasizes the retention of performance with fewer accumulation bits compared to other FQT methods, the hardware evaluation is conducted against hardware using 16-bit activation/accumulation. This makes it difficult to intuitively understand how reducing the accumulation of bits benefits the hardware.

[1] Xi et al., Training Transformers with 4-bit Integers
[2] Wang et al., Training Deep Neural Networks with 8-bit Floating Point Numbers

**Questions:**

- What are particular characteristics of CIL (distinct from NoCL) that require innovation for existing FQT?
- Is it possible to achieve a practical speed-up through the proposed method? Additionally, I'm curious about the overhead incurred by Hadamard transformation.
- I'm curious about the hardware benefits of using 8-bit or 12-bit accumulation with 4-bit input compared to 16-bit accumulation.
- In Table 2, HDQT in BiC exhibits both high accuracy and a high forget score. I'm curious if this is simply a result of overtraining the model or if it has achieved better performance than other methods.
- As depicted in Figure 1, it seems that the input and weight undergo a process of quantization-transformation-quantization when used in the backward pass. I wonder about the impact of double quantization like this.

---

> ### Author Response · Authors · 2023-11-21
>
> We would like to express our gratitude for the thorough review of our paper. We appreciate the insightful feedback provided by the reviewers. In response, we have carefully addressed each concern and question. In the paper, you can identify changes attributed to your reviews by R1.
>
> R1: This paper lacks novelty. While it introduces FQT to CIL, it is unclear whether the proposed method offers something significantly new compared to the existing FQT with Hadamard transform [1]. Also, the explanation regarding one of the paper's contributions, tiled matrix multiplication, does not provide sufficient details such as hyperparameters like tile size, which are essential for both accuracy and hardware cost. Additionally, the paper does not explain any distinctive aspects compared to chunk-based accumulation proposed in other existing FQT methods [2].
>
> Authors: We respectfully disagree with the reviewer that our paper lacks novelty and we would like to highlight that to the best of our knowledge, we are the first to apply fully quantized training to continual learning and to develop a quantization method which outperforms SOTA FQT methods in the CL setting. Our method is novel and different from the previous method [1] in that we apply Hadamard transform to backward pass to accommodate for gradient distributions and learning dynamics of CL. In [1],  Hadamard transform was applied to account for outliers in the forward pass activations. [1] does not explicitly use Hadamards transform in the backward pass, while they propose an alternative method for the backward pass using bit-splitting which leverages sparsity. We furthermore identify pain points such as accumulator bit-width in the forward pass (note that it is not necessary in the backward pass) and present remedies to recover substantial amounts of accuracy. We have updated the manuscript to include the results of a hyper-parameter study of tiled size in Table 4 in appendix A6 to demonstrate how we picked the tile size in the paper. We find that tile size 32 is the sweetspot as any larger shows performance degradation and increases in terms of standard deviation. Additionally, we have added a reference to [2] in the paper and we would like to highlight that [2] is proposing a blanket chunk based method applied to all matmuls in the network, we however provide an analysis and refinement that determines that for our type of workloads we only need a chunk/tile based accumulator matmul in the forward pass.
>
> R1: There is a lack of detailed analysis regarding the overhead incurred by Hadamard transformations in the training process. Furthermore, in [1], Hadamard transformation was used effectively to manage outliers occurring in specific channels of the Transformer model. However, this paper lacks a clear explanation of why the Hadamard transformation is particularly necessary for CIL. In Section 3, the direct relevance of being 'more efficient in using available quantization levels' to performance is not well-established.
>
> Authors: The Hadamard transform has a theoretical time complexity of O(m log m) and we also include the Hadamard transform in the energy numbers we provide when we use our quantization scheme. We can see that given our energy numbers we still materialize substantial gains compared to floating point baseline. We have modified our text to reflect the overhead of Hadamard transform better now. Additionally, we would like to point out that the choice for Hadamard transform is based on our observation of the gradient distribution during continual learning (see Fig. 2 which was derived from our logs during experiments) and previous literature analyzing general gradient distributions [3,4]. Due to its spreading properties and efficient implementation, hardware computation of Hadamard transforms have seen widespread use in spread-spectrum receivers such as those used in cellphones (code division multiple access CDMA) [6]. Building on this insight and the work shown in [4] we apply the Hadamard transform in the backward pass rather than the forward pass to more evenly spread information across quantizer levels (sometimes to the degree of using completely unused quantizer levels, see Fig. 2). Additionally, both intuitively and empirically increasung the number of quantizer levels used does improve performance, e.g. LUQ performance improved when using 17 vs 16 quantizer levels (Table 2 LUQ (17 levels) vs corrected LUQ (16 levels)). We have updated the text to reflect this and give a clearer explanation.

---

> ### Author Response · Authors · 2023-11-21
>
> R1: The rationale behind using stochastic rounding is unclear. In Figure 2 (right), the sensitivity of activation and gradient quantization in the backward pass to unbiased quantizers lacks a clear bridge of explanation. There is a lack of description regarding which factors among bias and variance make gradient and activation quantization in the backward pass challenging.
>
> Authors: Stochastic rounding is an established method to combat gradient bias in several previous works [4, 5, 7]. We additionally added figure 8 in appendix A13 to show the bias and variance development w/wo stochastic rounding in comparison to no quantization, highlighting the importance of stochastic rounding which is selectively applied (see Fig. 1 where we apply it and Fig. 2 that motivates our choice). This provides further evidence in addition to prior work [11] that stochastic rounding for FQT in CL results in bias and variance in gradient estimation closer to the unquatized version.
>
>
> R1: The hardware comparison is unfair. While this paper emphasizes the retention of performance with fewer accumulation bits compared to other FQT methods, the hardware evaluation is conducted against hardware using 16-bit activation/accumulation. This makes it difficult to intuitively understand how reducing the accumulation of bits benefits the hardware.
>
> Authors: We have added energy estimation for 12-bit and 16-bits accumulator bit width with and without Hadamard overhead to demonstrate a fairer hardware comparison. As expected the scaling behaves as shown in previous works [8,9]. In our case the progression gives increasing energy savings 3.3x (INT16) -> 3.4x (INT12) -> 4.5x (INT8) to for the hidden layer. Therefore, we can see significant benefits due to the usage of 8-bit accumulators.
>
> [1] Xi, Haocheng, et al. "Training Transformers with 4-bit Integers." arXiv preprint arXiv:2306.11987 (2023)
> [2] Wang, Naigang, et al. "Training deep neural networks with 8-bit floating point numbers." Advances in neural information processing systems 31 (2018).
> [3] Chmiel, Brian, et al. "Neural gradients are near-lognormal: improved quantized and sparse training." arXiv preprint arXiv:2006.08173 (2020).
> [4] Chmiel, Brian, et al. "Logarithmic unbiased quantization: Practical 4-bit training in deep learning." (2021).
> [5] Xia, Lu, et al. "A simple and efficient stochastic rounding method for training neural networks in low precision." arXiv preprint arXiv:2103.13445 (2021).
> [6] Glas, Jacobus Petrus Franciscus. "Non-cellular wireless communication systems." (1998): 1090-1090.
> [7] Sun, Xiao, et al. "Ultra-low precision 4-bit training of deep neural networks." Advances in Neural Information Processing Systems 33 (2020): 1796-1807.
> [8] V. Camus, L. Mei, C. Enz and M. Verhelst, "Review and Benchmarking of Precision-Scalable Multiply-Accumulate Unit Architectures for Embedded Neural-Network Processing," in IEEE Journal on Emerging and Selected Topics in Circuits and Systems, vol. 9, no. 4, pp. 697-711, Dec. 2019, doi: 10.1109/JETCAS.2019.2950386.
> [9] Murmann, Boris. "Mixed-signal computing for deep neural network inference." IEEE Transactions on Very Large Scale Integration (VLSI) Systems 29.1 (2020): 3-13.
> [11]Chang et al. “Rethinking the importance of quantization bias, toward full low-bit train-
> ing” IEEE Transactions on Image Processing, 31:7006–7019, 2022.

---

> ### Author Response · Authors · 2023-11-21
>
> Questions:
>
> R1: What are particular characteristics of CIL (distinct from NoCL) that require innovation for existing FQT?
>
> Authors: Our technique is advantageous for continual learning for the following reasons:
> - In Figure 2, we show that the gradient distributions encountered during the CL learning process are not favourable for uniform integer quantization. We empirically demonstrate that this is mitigated through the use of the spreading properties of the Hadamard transform.
> - In the new figures 9 and 10 we highlight that HDQT is better at approximating gradient distributions at lower bit widths.
> - We use an adaptive batch dependant quantization range to deal with the changing data distributions from CL. Our simplicity is an advantage for CL because we have no need for data samples or other parameter storage. This allows us to preserve resources for the often intensive CL techniques[1,2].
>
> [1] Hayes and Kanan “Online Continual Learning for Embedded Devices” (2022)
>
> [2] Verwimp et al. "Continual Learning: Applications and the Road Forward" arXiv preprint arXiv:2311.11908 (2023)
>
> R1: Is it possible to achieve a practical speed-up through the proposed method? Additionally, I'm curious about the overhead incurred by Hadamard transformation.
>
> Authors: We have added additional numbers in Table X for estimates without the Hadamard transform and we see roughly a 17.7x decrease in energy for using the transform. We want to highlight that we use an unoptimised worst-case implementation of the Hadamard transform (n^2 multiplies rather than n log n adds) due to the limitations of the architectural performance estimations software. In practice, we anticipate custom implementations that could dramatically improve these results. Our efforts towards implementing these techniques using reconfigurable hardware could not be included in the current manuscript due to time and space constraints.
>
> R1: I'm curious about the hardware benefits of using 8-bit or 12-bit accumulation with 4-bit input compared to 16-bit accumulation.
>
> Authors: We have added energy estimation for 12-bit and 16-bits accumulator bit width with and without Hadamard overhead to demonstrate a fairer hardware comparison. As expected the scaling behaves as shown in previous works [8,9]. In our case the progression of gives increasing energy savings 3.3x (INT16) -> 3.4x (INT12) -> 4.5x (INT8) to for the hidden layer. Therefore, we can see significant benefits due to the usage of 8-bit accumulators.
>
> R1: In Table 2, HDQT in BiC exhibits both high accuracy and a high forget score. I'm curious if this is simply a result of overtraining the model or if it has achieved better performance than other methods
>
> Authors: The reason for a high accuracy together with a high forgetting score is that the forgetting score is simply the average difference of the class accuracy to their best-achieved performance so far. From this, we can infer that in the case of BIC, our method is more susceptible to incorporating new data and achieving higher intermittent class accuracies.
>
> R1: As depicted in Figure 1, it seems that the input and weight undergo a process of quantization-transformation-quantization when used in the backward pass. I wonder about the impact of double quantization like this.
>
> Authors: We have added section 7 in the appendix where we include a quality study without the second re-quantization. We find that there is no significant difference with and without the re-quantization but want to highlight that the Hadamard transform might result in values requiring a higher precision representation hence it is necessary to re-quantize to guarantee our desired input bit-precision for the matrix multiplication. Further, this re-quantization can be directly embedded in the hardware calculating the Hadamard transform with negligible overhead.

---

### Meta-Review · Area_Chair_pXJF · 2023-12-17

**Metareview:**

The paper develops a fully quantized training method for an incremental learning setting.  The underlying technical approach relies upon Hadamard transforms followed by quantization or stochastic quantization operations in the backward pass; these building blocks have been developed in prior work --- e.g., [Xi et al., Training transformers with 4-bit integers, 2023] proposes the Hadamard quantizer.

Reviewers unanimously lean toward reject, with a common concern that there is minimal innovation in terms of fundamental techniques for neural network quantization (Reviewer ZGvQ: "unclear whether the proposed method offers something significantly new compared to the existing FQT"; Reviewer 2d5P: "previous work that applies the Hadamard transform to quantization").  Reviewers YCXn and ZGvQ also ask for additional experiments or analysis.

The AC has read the author response.  This response does address some reviewer questions and articulates that the authors view the application to continual learning as the contribution.  However, the AC agrees with the reviewers and remains concerned that the overall significance of this contribution is not sufficient.

**Justification For Why Not Higher Score:**

While the paper demonstrates quantization techniques applied to the continual learning setting, its contributions towards advancing fundamental techniques for neural network quantization are unclear.

**Justification For Why Not Lower Score:**

N/A

---

### Decision · Program_Chairs · 2024-01-16

Reject